# Transcriptome Sequencing Reveals the Mechanism of Auxin Regulation during Root Expansion in Carrot

**DOI:** 10.3390/ijms25063425

**Published:** 2024-03-18

**Authors:** Xuan Li, Xuemin Yan, Zhe Wu, Leiping Hou, Meilan Li

**Affiliations:** College of Horticulture, Shanxi Agricultural University, Jinzhong 030801, China; lx20130505130@163.com (X.L.); yanxuemin0120@163.com (X.Y.); wzz0618@163.com (Z.W.); sxndhlp@126.com (L.H.)

**Keywords:** carrot, root expansion, IAA, transcriptome sequencing, gene

## Abstract

Carrot is an important vegetable with roots as the edible organ. A complex regulatory network controls root growth, in which auxin is one of the key players. To clarify the molecular mechanism on auxin regulating carrot root expansion, the growth process and the indole-3-acetic acid (IAA) content in the roots were measured in this experiment. It was found that the rapid expansion period of the root was from 34 to 41 days after sowing and the IAA content was the highest during this period. The root growth then slowed down and the IAA levels decreased. Using the transcriptome sequencing database, we analyzed the expression of IAA-metabolism-related genes and found that the expression of most of the IAA synthesis genes, catabolism genes, and genes related to signal transduction was consistent with the changes in IAA content during root expansion. Among them, a total of 31 differentially expressed genes (DEGs) were identified, including 10 IAA synthesis genes, 8 degradation genes, and 13 genes related to signal transduction. Analysis of the correlations between the DEGs and IAA levels showed that the following genes were closely related to root development: three synthesis genes, *YUCCA10* (*DCAR*_*012429*), *TAR2* (*DCAR*_*026162*), and *AMI1* (*DCAR*_*003244*); two degradation genes, *LPD1* (*DCAR*_*023341*) and *AACT1* (*DCAR*_*010070*); and five genes related to signal transduction, *IAA22* (*DCAR*_*012516*), *IAA13* (*DCAR*_*012591*), *IAA27* (*DCAR*_*023070*), *IAA14* (*DCAR*_*027269*), and *IAA7* (*DCAR*_*030713*). These results provide a reference for future studies on the mechanism of root expansion in carrots.

## 1. Introduction

Carrot (*Daucus carota* var. *sativa* DC.) is a biennial plant with nutritious fleshy roots that are widely consumed. Rich in carotenoids, vitamins, lignin, and a variety of other nutritional elements, carrots can enhance immunity, prevent cancers and other diseases, and lower cholesterol in humans [1]. Since roots are important edible organs in carrots, it is of great significance to study root expansion.

Fleshy root expansion is a complex process and includes morphogenesis and dry matter accumulation. It is influenced and regulated by multiple factors, including environmental factors and endogenous hormones [2]. Previous studies have shown that the endogenous hormones in plants mainly affect the growth and development of the roots by controlling the division and differentiation of root cells; hence, they play an important role in root expansion [3]. Auxin, the first type of plant hormone discovered, is widely distributed in plants but mostly concentrated in root tip meristem, cambium, and young seeds. This hormone regulates plant growth and development, cell elongation, and division and promotes xylem and phloem differentiation [4]. The formation of sweet potato root tubers is caused by auxin-induced activation of the root cambium and xylem cell proliferation, thus promoting the enlargement of the root tubers [5]. Moreover, IAA participates in the formation of lateral roots and the change in root architecture of sweet potato under the low K^+^ stress [6].

Root development is closely related to the activity of secondary cambium, which is determined by plant hormones. Studies have shown that indole-3-acetic acid (IAA) exists in a dynamic balance in plants, and its content and distribution in the roots directly determine cell division [7]. In Zingiber, the levels of the hormones such as IAA, gibberellic acid (GA), and jasmonic acid (JA) were significantly increased with increasing rhizome diameter [8]. However, the role of IAA in root formation remains contentious. Some researchers believe that IAA affects the activity of lignin and regulates its synthesis in carrot, thus affecting root formation [9,10].

In this study, the carrot roots were investigated at five growth stages. By combining the IAA content and transcriptome sequencing (RNA-Seq) results, the mechanism of IAA regulation expansion was assessed from multiple perspectives. The findings lay the theoretical foundation for the study of carrot root expansion mechanisms in the future, with a view to providing valuable reference data for high-yield breeding of other root crops.

## 2. Results

### 2.1. Analysis of the Carrot Root Morphological Indices

The root length and diameter of the carrots gradually increased after sowing (Figure 1A,B). At the early stages, it was mainly the root length that increased, and the rate peaked at 34 days after sowing, reaching 53.6%; however, this decreased rapidly at 41 days after sowing and maintained flat thereafter (Figure 1C). The increase in the root diameter began 34 days after sowing and the growth rate increased rapidly from 34 to 41 days, reaching 130.5% before slowing down (Figure 1D).

### 2.2. Comparison of the IAA Content in Carrot Roots at Different Growth Stages

The content of IAA in the carrot roots was measured at different time points (Figure 2A). The IAA content was higher 34 and 41 days after sowing and peaked at 41 days, followed by a gradual decrease. The IAA content differed significantly between 34 and 41 and between 41 and 48 days. Additionally, the correlation analysis between the growth indicators and IAA content showed that IAA content was negatively correlated with root length (−0.70) and root diameter (−0.83), indicating that IAA is associated with root expansion (Figure 2B).

### 2.3. Analysis of Transcriptome Sequencing

#### 2.3.1. Quality Assessment of Transcriptome Sequencing Data

A total of 15 cDNA libraries were constructed in order to analyze the gene expression in the carrot roots at different growth stages. After filtering out the contaminating sequences and low-quality sequences from the raw data, the high-quality reads were all in the range of 91.13–94.52% and of a good sequencing quality (Appendix A). The efficiency of the high-quality reads obtained from each library against the carrot reference genome (http://plants.ensembl.org/Daucus_carota/Info/Index (accessed on 14 January 2022)) ranged from 86.97 to 88.41%, confirming the reliability of the data and their suitability for subsequent experimental analysis.

#### 2.3.2. Sample Repeatability Test

The repeatability between biological replicates reflects the reliability of the data. Principal component analysis provides a visual demonstration of whether the biologically replicated samples are sufficiently similar and whether the differences between groups are sufficiently large. Samples of the carrot roots from different growth stages were clustered together and clearly distinguished from other stages, suggesting that the differences within the groups were small and the differences between the groups were significant (Appendix A). Spearman’s correlation coefficient used to evaluate the correlation between the biological replicates of all samples (Appendix A) further confirmed the robustness of the sequencing data.

#### 2.3.3. Real-Time Quantitative PCR Validation of the RNA-Seq Data

Six genes were randomly selected for real-time quantitative PCR (RT-qPCR) to verify the reliability of the gene expression results obtained with RNA-Seq. As shown in Appendix A, the fragments per kilobase of exon model per million mapped fragments (FPKM) values of *DCAR_012429*, *DCAR*_*030523*, *DCAR*_*003708*, *DCAR*_*026162*, *DCAR*_*016234*, and *DCAR*_*018164* were consistent with the fluorescence quantitative results, indicating that the transcriptome data were reliable.

### 2.4. Expression Analysis of IAA-Metabolism-Related Genes

Studies have shown that the aromatic amino acid l-tryptophan (Trp) is the central precursor for IAA biosynthesis in plants [11]. Trp-dependent IAA biosynthesis involves multiple parallel pathways, leading to the synthesis of IAA as the final product (Figure 3). The indole-3-pyruvic acid (IPyA) pathway is the main pathway for the synthesis of IAA in plants and consists of a two-step reaction in which Trp is first deamidated by tryptophan aminotransferase of arabidopsis 1 (TAA1) into IPyA. It is then decarboxylated into IAA and catalyzed by a flavin-containing monooxygenase of the YUCCA (YUC) family [12]. The Trp pathway and the IPyA pathway share the same first step, and the intermediate product, IPyA, is decarboxylated into IAA in the presence of the key enzymes aldehyde dehydrogenase (ALDH, EC: 1.2.1.3) and indole-3-acetaldehyde oxidase (AAO, EC: 1.2.3.7). In the indole-3-acetaldoxime (IAOx) pathway, the conversion of Trp into its derivative IAOx is catalyzed by the cytochrome P450 monooxygenases CYP79B2 and CYP79B3 [13]. Indole-3-acetonitrile (IAN) is then finally converted into IAA by the action of the plant nitrilases family (NITs) [14]. In addition, Trp generates the intermediate product indole-3-acetamide (IAM) via the IAM pathway, which, in turn, catalyzes the conversion of IAM into IAA by AMIDASE1 (AMI1) [15,16].

The main catabolic pathway regulating the IAA levels in plants is the irreversible oxidation of IAA into 2-oxoindole-3-acetic acid (oxIAA) by protein dioxygenase for auxin oxidation (DAO) [17]. In addition, intermediates of the IAA degradation pathway are involved in metabolic activities such as aminobenzoic acid catabolism, nicotinamide metabolism, benzoic acid catabolism, and glycolysis. The key enzymes involved in the catabolic pathway are catalase (CAT, EC: 1.11.1.6), 2-oxoglutarate dehydrogenase (OADH, EC: 1.2.4.2), enoyl-CoA hydratase (echA, EC: 4.2.1.17), and acetyl-CoA C-acetyltransferase (ACAT, EC: 2.3.1.9) [18].

#### 2.4.1. Expression Analysis of IAA Synthesis Genes

The synthesis of IAA in plants is divided into four pathways: the tryptamine pathway, the IPyA pathway, the IAOx pathway, and the IAM pathway (Figure 3). Analysis of the genes encoding the key enzymes in the IAA synthesis pathway identified 49 genes involved in the process (Table 1). Most of the genes encoding TAA1, a key enzyme of the tryptamine and IPyA pathways, were expressed at higher levels at the early growth stages than at later growth stages, which coincided with the elevated IAA content during early growth. Subsequently, the gene expression level decreased, as did the IAA content, resulting in slower carrot root expansion. Within this, the expression of *DCAR*_*026162* varied significantly during the root expansion. In the tryptamine pathway, the expression changes in most of the genes encoding the key enzyme ALDH were consistent with the changes in the IAA content. Among them, *DCAR*_*011215*, *DCAR*_*024255*, *DCAR*_*031309*, and *DCAR*_*009596* were significantly differentially expressed during root expansion. In the IPyA pathway, there were 19 genes encoding YUC [19], of which *DCAR*_*012429* and *DCAR*_*000740* were significantly differentially expressed during root expansion. In the IAM pathway, eight genes encoding AMI1 had higher expression levels at the early growth stages than the later growth stages. Of these, *DCAR*_*003244* and *DCAR*_*003706* were significantly up-regulated during root expansion, which was consistent with the IAA content. These genes play a crucial role in IAA synthesis and are responsible for the changes in IAA content. However, the expression of IAOx-pathway-related genes was not detected, suggesting that the IAOx pathway may not be involved in the process of carrot root expansion. Therefore, we propose that the main IAA synthesis pathways in carrot root expansion are the tryptamine, IPyA, and IAM pathways.

#### 2.4.2. Expression Analysis of Auxin Degradation Genes

IAA catabolism includes aminobenzoate degradation, nicotinamide metabolism, benzoate degradation, and glycolysis. The key enzymes involved in the IAA catabolic pathway are shown in Figure 3. Analysis of expression of the genes encoding these key enzymes revealed that a total of 31 genes were involved in the IAA catabolic process (Appendix A). The DEGs and some genes with annotation of the most likely orthologue genes in the degradation and their expressions level are shown in Table 2. Among these, *DCAR_013925* and *DCAR_009536*, which encode CAT, were significantly down-regulated in H34 vs. H41, which is the opposite trend to that detected for the IAA content. The DAO-encoding genes were all down-regulated during root expansion, among which *DCAR_016234* and *DCAR_016235* were significantly down-regulated. This is consistent with the results of Porco et al., which suggested that *AtDAO1* in *Arabidopsis* regulates the length of the root hairs and controls the auxin levels in vivo [20]. In this experiment, four DAO-encoding genes (*DCAR_016234*, *DCAR_002760*, *DCAR_006175*, and *DCAR_006176*) were up-regulated in H41 vs. H48 to maintain lower IAA content, required for carrot root growth in a suitable range. *DCAR_016716* and *DCAR_022577*, *DCAR_024929*, *DCAR_024930*, *DCAR_002529*, *DCAR_026968*, and *DCAR_029358*, which encode OADH, another key enzyme in the catabolic pathway of IAA, and *DCAR_019674*, *DCAR_006315*, *DCAR_010070*, and *Daucus_carota_newGene_5365*, which encode ACAT, were down-regulated during root expansion, which is consistent with the changes in IAA content. This suggests that they may be involved in glycolysis and decompose excess auxin in the carrot root to maintain auxin concentration homeostasis.

#### 2.4.3. Expression Analysis of Genes Related to Auxin Signal Transduction

The response to auxin in plants is mainly controlled by auxin signal transduction. Typically, the auxin signal transduction pathway is composed of the responses of transport inhibitor response 1 (TIR1), auxin/indole-3-acetic acid (Aux/IAA), and auxin response factor (ARF) (Figure 4). Among these, the TIR1 protein is an important component of the SCF^TIR1^ complex, which is considered an auxin receptor, and its mutant tir1 exhibits defective hypocotyl and lateral root formation [21]. Aux/IAA is a class of auxin-responsive proteins. At low auxin concentrations, the SCF^TIR1^ complex interacts with Aux/IAA to form the Aux/IAA-ARF transcriptional regulatory element. When its concentration is elevated, auxin mediates the Aux/IAA proteins to undergo ubiquitination degradation, thereby releasing ARF, which promotes the transcription of the corresponding genes and regulates plant growth and development [22].

A total of 78 genes were identified to be involved in the transcriptional regulation of the three proteins TIR1, Aux/IAA, and ARF in this experiment (Appendix A) and the DEGs and key genes are shown in Table 3. The expression of *DCAR_028134*, the gene encoding TIR1, was significantly higher at the early growth stages (34 d and 41 d after sowing) than at the later growth stages (48 d, 55 d, and 62 d after sowing), which was consistent with the trend in IAA content. There were 33 AUX/IAA-encoding genes identified (Appendix A, Appendix A), which were more sensitive to the auxin concentration. The expression of 14 genes (*Daucus_carota_newGene_6588*, *DCAR_010763*, *DCAR*_*012516*, *DCAR*_*012591*, *DCAR*_*018241*, *DCAR*_*018447*, *DCAR*_*002354*, *DCAR*_*023070*, *DCAR*_*024202*, *DCAR*_*026353*, *DCAR*_*027352*, *DCAR*_*027269*, *DCAR*_*030712*, and *DCAR*_*030713*) was significantly higher at the early growth stages than the later growth stages, consistent with the trend in the change in IAA content. This indicates that it might be the AUX/IAA proteins that responded to the changes in IAA content and regulated the expression of the transcription factors in the roots. ARF is a key transcription factor in the auxin signal transduction pathway and works together with the AUX/IAA proteins to regulate the expression of auxin-related genes [23]. *DCAR*_*015554*, *DCAR*_*014788*, *DCAR*_*017496*, *DCAR*_*020464*, *DCAR*_*005400*, and *DCAR*_*006053*, which encode ARF, showed significant differences in expression, and all of them were higher at the early growth stages. The expression of the six ARF genes was consistent with the trend in the change in IAA content. In *Arabidopsis*, *AtARF5*, *AtARF6*, *AtARF7*, *AtARF8*, and *AtARF19* together form a transcriptional activator that acts on flower, seedling, and root development as well as hypocotyl elongation [24], and the double mutants of *arf19* and *arf7* exhibit auxin resistance [25], suggesting that the ARF transcription factor controls the auxin response during carrot root expansion.

### 2.5. Discovery of Genes Related to IAA Regulation during Root Expansion

Screening of the differentially expressed genes (DEGs) with false discovery rate (FDR) < 0.01 and fold change (FC) ≥ 1.5 found a total of eight IAA synthesis genes, seven catabolism genes, and twelve genes related to auxin signal transduction that were differentially expressed during root expansion (Table 4). In order to obtain the key genes for auxin regulation of the root expansion, all of the above DEGs were correlated with the IAA content and a total of 11 DEGs with correlation coefficients ≥ 0.8 were identified (Figure 5). Their expression was significantly correlated with the changes in IAA content. In addition, by analyzing the expression patterns of the genes in the IAA metabolic pathways, we found three synthetic genes, *YUCCA10* (*DCAR*_*012429*), *TAR2* (*DCAR*_*026162*), and *AMI1* (*DCAR*_*003244*); one catabolic gene *AACT1* (*DCAR*_*010070*); and five signaling genes *IAA22* (*DCAR*_*012516*), *IAA13* (*DCAR*_*012591*), *IAA27* (*DCAR*_*023070*), *IAA14* (*DCAR*_*027269*), and *IAA7* (*DCAR*_*030713*) were differentially expressed in H41 vs. H48, and one catabolic gene *LPD1* (*DCAR* _*023341*) was differentially expressed in H55 vs. H62 (Table 5), and these genes were consistent with the expression of genes involved in IAA metabolism in the above analysis.

*DCAR_012429*, *DCAR*_*026162*, and *DCAR*_*003244* encode the YUCCA, TAA1, and AMI1 enzymes, respectively, which are involved in the tryptamine, IPyA, and IAM pathways of IAA synthesis. Expression of these genes was down-regulated in H41 vs. H48, indicating that the enzyme activity decreased, and the IAA content gradually decreased on 48 d after sowing. These results are consistent with the observed changes in the IAA content, indicating that the down-regulated gene expression led to a decrease in the IAA content and the root expansion rate consequently decreased gradually. Expression levels of *DCAR*_*012429* and *DCAR*_*003244* were up-regulated in H34 vs. H41, indicating that the rate of IAA synthesis increased and the IAA content also increased from 34 d to 41 d after sowing. At this stage, the carrot roots were in a phase of rapid growth. *DCAR*_*010070* and *DCAR*_*023341* encode the ACAT and OADH enzymes, respectively, and they were up-regulated in H41 vs. H48 and in H55 vs. H62. Acetyl-CoA was continuously produced to participate in glucose metabolism and maintain the dynamic IAA balance in the plants. The DEGs involved in auxin signal transduction were all genes encoding the AUX/IAA protein and were down-regulated in H41 vs. H48. The activities of the AUX/IAA-encoding genes were affected at once when the IAA content changed, indicating that the AUX/IAA transcription factor family regulated the synthesis of IAA from 41 d to 48 d after sowing and led to a drastic decrease in the IAA content (Table 5 and Figure 6). Therefore, it is suggested that the above 11 genes are candidate genes that play a crucial role in IAA regulation of carrot root expansion, but their specific mechanisms of action need to be further explored.

## 3. Discussion

During the root formation of carrot, changes in endogenous hormone content are the main regulatory factors. The process of maintaining continuous root growth involves a complex network of hormonal regulation, in which auxin plays an important role. High concentrations of auxin in the roots have the effect of localizing and establishing root stem cell populations, which can promote cell differentiation, whereas the process of the repeated division of daughter cells requires appropriately reduced auxin concentrations [26]. In the present study, the IAA content at the early growth stages of root development was much higher than at the later growth stages, which confirms that auxin plays an important role in root initiation and formation processes. However, Perrin et al. showed that carotenoids are differentially accumulated in phloem and xylem carrot root depending on the genotype [27]. Therefore, whether there are differences in auxin content between different tissues in carrot root and its specific mechanism will be a valuable research direction. In addition, Khadr et al. showed that increased IAA content may cause an increase in the number of xylem cells [28]; however, the specific mechanism needs to be further studied.

To further clarify the role of IAA in the root expansion of carrots, we analyzed the associated metabolic processes. No genes related to the IAOx pathway were detected, suggesting that this pathway may not exist in carrot root growth. Studies have shown that IAOx plays a role in the synthesis of IAA in *Arabidopsis thaliana* and this pathway is specific to *Brassica* crops [29,30].

In the tryptamine and the IPyA pathways, *TAR2* (*DCAR*_*026162*) encodes L-tryptophan-pyruvate transaminase, which mainly acts on the elongation of the hypocotyls, lateral root formation, gravity growth [31], and conversion of tryptophan into indolepyruvate in plants. Subsequent research has confirmed that *TAA* and members of the *YUCCA* gene family are the first pathways jointly involved in IAA synthesis and the most important pathway in the IAA-dependent synthesis of tryptophan, controlling changes in IAA levels [12]. In the present study, the expression of the *YUCCA* gene *DCAR*_*012429* 41 days after sowing was significantly higher than at 48 days and regulated the rapid growth of the carrot roots, and it is a rate-limiting enzyme in the IPyA pathway. Therefore, we speculate that *YUCCA10* (*DCAR*_*012429*) is a candidate gene for IAA metabolism in the process of root expansion.

ALDH-encoding genes *ALDH2B4* (*DCAR*_*011215* and *DCAR*_*031309*) and *ALDH3H1* (*DCAR*_*009596*) are specifically expressed in carrot roots, and they mainly respond to abiotic stress. They are considered plant antidotes [32]. AMI1, a key enzyme in the IAM pathway, hydrolyzes indole-3-acetamide to produce IAA, and this specific amidase is widespread among plants. Genes encoding this enzyme are actively expressed in tissues with a high IAA content, which, in plants, are mostly vigorously growing tissues [33,34], consistent with our results. *DCAR_003708* and *DCAR_003244* were highly expressed alongside elevated IAA levels at the early growth and developmental stages in the carrots. In IAA degradation, CAT, DAO, OADH, and ACAT are key enzymes that metabolize excess IAA and maintain the IAA balance in plants, thereby promoting root expansion [35].

IAA is synthesized in plants and specifically binds to the auxin receptor SCF^TIR1^ in the nucleus, thus starting the auxin signal transduction pathway. Recently, a study showed that TIR1/AFB-Aux/IAA pathway rapidly modulates root cell elongation [36]. In our experiment, only the AUX/IAA-encoding genes are differentially expressed, as they are more sensitive to changes in IAA content. When the expression of the gene *YUCCA10* (*DCAR*_*012429*) was up-regulated in H34 vs. H41, resulting in an increase in IAA content, signal transduction was actively expressed in the plant and the corresponding genes were up-regulated, thereby promoting root expansion.

## 4. Materials and Methods

### 4.1. Plant Materials and Cultivation

The ‘Dongfanghong’ carrot variety was selected as the experimental material and coated carrot seeds were sown in a plastic greenhouse at the experimental station of the College of Horticulture, Shanxi Agricultural University, China, on 7 August 2020. Ridge cultivation was employed with a width of 60 cm and a height of 20 cm. Two rows were sown on each ridge and the carrot plants were traditionally managed.

### 4.2. Phenotyping and Sampling of the Carrot Roots at Different Growth Stages

#### 4.2.1. Determination of Morphological Indices in the Carrot Roots

To understand the dynamics of the carrots during enlargement, the root length and diameter were measured weekly from 13 to 62 days after sowing.

#### 4.2.2. Sample Collection

Carrots were harvested 34, 41, 48, 55, and 62 days after sowing. Fresh roots were washed and the middle part of the carrot roots with similar growth (1/3 to 2/3 from the growing point) was sampled in triplicate. The samples were named H34-1, H34-2, H34-3, H41-1, H41-2, H41-3, H48-1, H48-2, H48-3, H55-1, H55-2, H55-3, H62-1, H62-2, and H62-3, respectively. Samples were placed in a centrifuge tube, flash-frozen in liquid nitrogen, and stored at −80 °C until use.

#### 4.2.3. Determination of the IAA Content at Different Growth Stages

Approximately 2 g of each sample was taken in triplicate and IAA content was determined using high-performance liquid chromatography (HPLC), as described by Zhong et al. [37], using a Hypersil ODS C18 column (250 mm × 4.6 mm, 5 μm; Thermo Scientific, Waltham, MA, USA) and an Agilent 1200 UPLC system (Agilent Technologies Inc., Santa Clara, CA, USA). Isocratic elution was performed using a mobile phase of 20:80 methanol/acetic acid aqueous solution (pH 3.6) at a flow rate of 1 mL·min^−1^ with an injection volume of 20 μL. The detection wavelength was 254 nm.

### 4.3. Transcriptome Sequencing of Carrot Roots at Different Growth Stages

#### 4.3.1. RNA Extraction and Reverse Transcription

Approximately 1 g of each sample was taken in triplicate, an RNAprep Pure Plant Kit (DP432, TIANGEN, Beijing, China) was used to extract the total RNA, and the purity of the RNA was tested. Qualified RNA samples were added to magnetic beads with Oligo (dT) to enrich and fragment the mRNAs, which served as templates to synthesize and purify cDNAs. The cDNAs were then end-repaired, poly A tails added, and sequencing connectors attached. Appropriately sized fragments were selected for the construction of a cDNA library using PCR amplification. After quality testing (effective concentration of the library > 2 nM), the Illumina platform was used for sequencing.

#### 4.3.2. Analysis of the Transcriptome Sequencing Data

RNA-Seq was carried out by Biomarker Technologies Co., Ltd. (Beijing, China). The raw reads were further processed with a bioinformatic pipeline tool, BMKCloud (www.biocloud.net (accessed on 14 January 2022)) online platform. Clean data were obtained by filtering out the raw data containing connectors and reads of low quality and compared with the carrot reference genome (http://plants.ensembl.org/Daucus_carota/Info/Index (accessed on 14 January 2022)). Mapped reads were obtained by comparison using HISAT2 [38] and assembled by StringTie [39]. The original gene expression level was calculated by HTSseq and normalized with FPKM. Gene function was annotated based on the following databases: Gene Ontology (GO, https://geneontology.org/ (accessed on 14 January 2022)), Kyoto Encyclopedia of Genes and Genomes (KEGG, https://www.kegg.jp/ (accessed on 14 January 2022)), Ensemble Plants (https://plants.ensembl.org/Daucus_carota/Info/Index (accessed on 14 January 2024)), Panther (https://www.pantherdb.org/ (accessed on 14 January 2024)), and Dicots PLAZA 5.0 (https://bioinformatics.psb.ugent.be/plaza/versions/plaza_v5_dicots/ (accessed on 14 January 2024)). Plaza 5.0 was used to analyze the database integrated by InterPro, Mapman, and GO database [40]. DIAMOND version 2.0.4 software was used to blast the sequence of genes with the KEGG database [41], with a cut-off E-value 1 × 10^−5^. RNA-Seq data analysis was performed using BMKCloud platform.

RNA-Seq data from this study have been deposited into the NCBI SRA database under accession number PRJNA821411 (https://www.ncbi.nlm.nih.gov/sra/PRJNA821411 (accessed on 10 April 2022)). Corresponding sequence data are referenced to BioSample accessions SAMN27097857 (for H34), SAMN27097858 (for H41), SAMN27097859 (for H48), SAMN27097860 (for H55), and SAMN27097861 (for H62).

### 4.4. Analysis of IAA-Metabolism-Related Genes

From the RNA-Seq results, the expression of genes encoding key enzymes in the IAA synthesis and degradation pathways (tryptophan metabolism, ko00380) and auxin signal transduction pathway (plant hormone signal transduction, ko04075) in different periods was analyzed to explore the molecular basis of the IAA content changes during carrot root expansion. GO, KEGG, Ensemble Plants, Panther, and Dicots PLAZA database were used to annotate IAA metabolism and signal-transduction-related genes (Appendix A).

### 4.5. Screening of IAA-Metabolism-Related Genes during Root Expansion

The screening threshold for genes involved in IAA metabolism was set as FDR < 0.01 and FC ≥ 1.5, and the DEseq package was used to identify DEGs [42]. Expression levels of the DEGs and IAA levels were then correlation analyzed to obtain candidate genes related to IAA regulation in root expansion.

### 4.6. Correlation Analysis between IAA Content and IAA Metabolism and Signal-Transduction-Related DEGs

In order to obtain the key genes for auxin regulation during root expansion, with *p* ≤ 0.05 as the significant threshold, the correlation index was calculated by Pearson’s correlation method. The Gidio Bioinformatics Cloud Tool Platform (http://www.omicsshare.com/tools (accessed on 14 January 2024)) was used to perform the correlation analysis between IAA content and the FPKM values of DEGs related to IAA metabolism and signal transduction (Table 4).

### 4.7. Real-Time Quantitative PCR Analysis

Six DEGs (*DCAR_012429*, *DCAR*_*030523*, *DCAR*_*003708*, *DCAR*_*026162*, *DCAR*_*016234*, and *DCAR*_*018164*) were randomly selected for the RT-qPCR. *DcActin1* was selected as an internal reference gene and NCBI (https://www.ncbi.nlm.nih.gov/tools/primer-blast/index.cgi?LINK_LOC=BlastHome (accessed on 10 January 2023)) and the SnapGene version 6.2.2 software were used to design the primers (Appendix A).

cDNAs were synthesized using reverse transcription of the qualified RNA samples using a PrimeScriptTM RT reagent kit (Perfect Real Time, RR037A, TaKaRa, Beijing, China) and diluted to 80 ng·μL^−1^. An ABI 7500 fluorescence quantitative PCR instrument was used with TB Green^®^ Premix Ex Taq™ II (Tli RNaseH Plus, RR820A, Takara), and three biological replicates were included. Thermal cycling was performed at 94 °C for 5 min, followed by 40 cycles at 94 °C for 30 s, 56 °C for 30 s, and 72 °C for 30 s. Relative gene expression levels were calculated using the 2^−ΔΔCT^ method [43].

### 4.8. Statistical Analysis

Statistical analyses were performed using Microsoft Excel and SPSS version 28.0.1.1. Differences were analyzed by an ANOVA test, followed by the Duncan’s new multiple range test for root length and diameter of carrot and IAA content. Clustering of gene expression was performed using R version 3.6.1 pheatmap package with Ward method on Euclidian distances.

## 5. Conclusions

Dynamic analysis of the root growth in carrots showed that the rapid expansion period of roots was from 34 to 41 days after sowing. Determination of the IAA content at five different growth stages during root expansion revealed that the IAA content was the highest on 41 d, but the growth of the roots subsequently slowed down and the IAA levels also decreased. Analysis of the expression of genes encoding the key enzymes in IAA metabolism identified 21 IAA-synthesis-related genes, 10 IAA-degradation-related genes, and 22 genes related to auxin signal transduction that were consistent with the IAA content. Among them, 12 IAA synthesis DEGs, 8 IAA degradation DEGs, and 13 DEGs related to auxin signal transduction were obtained. Analysis of the correlations between the expression of these genes and the IAA levels showed that the following genes were most closely related to root expansion: three IAA synthesis genes, *YUCCA10* (*DCAR*_*012429*), *TAR2* (*DCAR*_*026162*), and *AMI1* (*DCAR*_*003244*); two degradation genes, *LPD1* (*DCAR*_*023341*) and *AACT1* (*DCAR*_*010070*); and five genes related to auxin signal transduction, *IAA22* (*DCAR*_*012516*), *IAA13* (*DCAR*_*012591*), *IAA27* (*DCAR*_*023070*), *IAA14* (*DCAR*_*027269*), and *IAA7* (*DCAR*_*030713*). These results provide a reference for a further exploration of the mechanism of root expansion in future work.

## Figures and Tables

**Figure 1 ijms-25-03425-f001:**
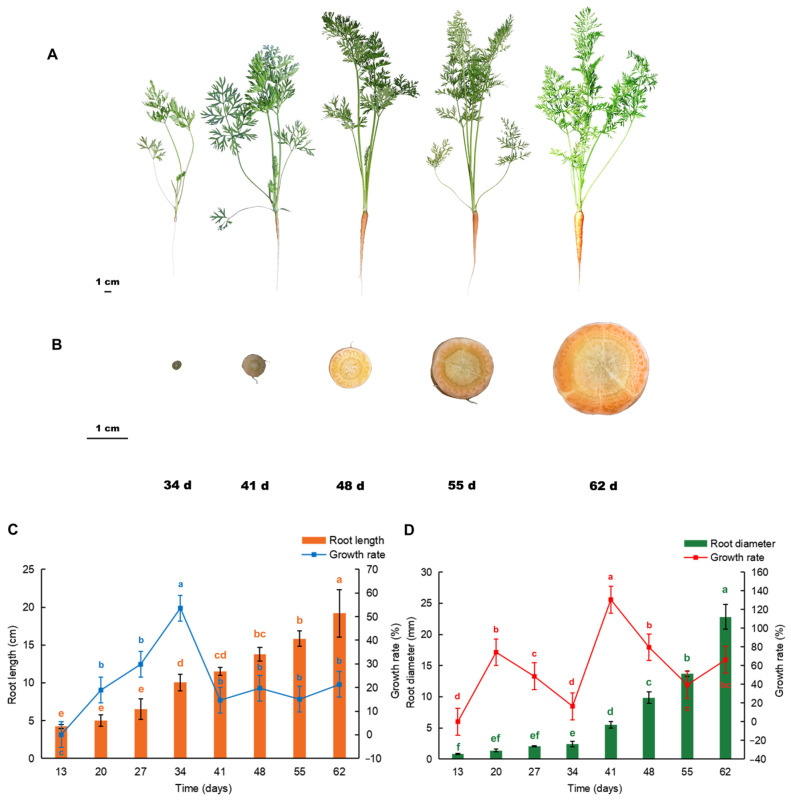
Root phenotype of carrot at different growth stages. (**A**) Appearance; (**B**) cross-section; 34 d, 41 d, 48 d, 55 d, and 62 d after sowing; (**C**) root length (cm) and growth rate; (**D**) root diameter (mm) and growth rate. Different letters indicate significant difference. Duncan’s test, *p* < 0.05.

**Figure 2 ijms-25-03425-f002:**
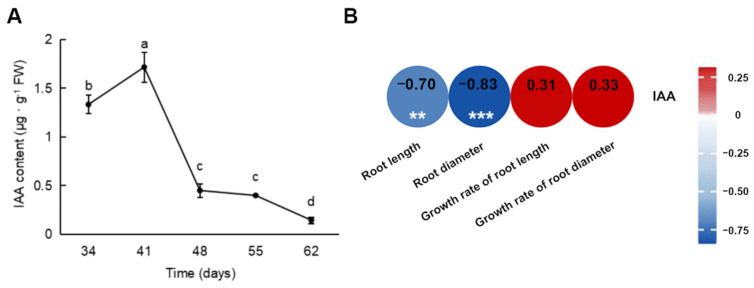
IAA content in carrot roots at different growth stages and heatmap of the correlation between the morphological indicators and IAA content. (**A**) IAA content; (**B**) heatmap of correlation analysis. ** *p* < 0.01, *** *p* < 0.001. Numbers: correlation coefficients.

**Figure 3 ijms-25-03425-f003:**
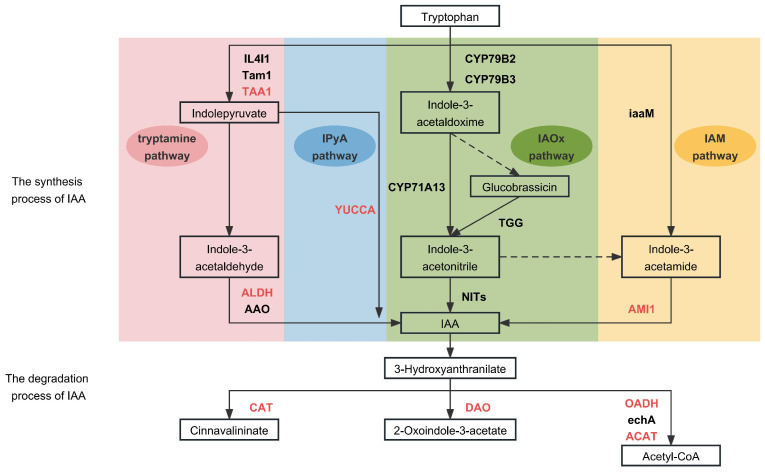
IAA metabolism process and key enzymes. The pink, blue, green, and yellow represent the tryptamine pathway, IPyA pathway, IAOx pathway, and IAM pathway, respectively. Red letters: key enzymes. Dotted lines: valid pathways.

**Figure 4 ijms-25-03425-f004:**
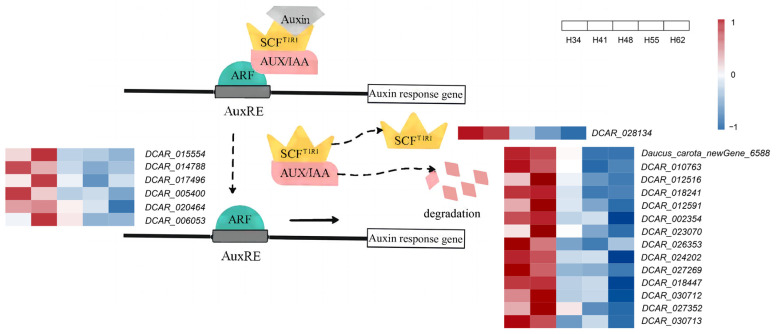
Auxin signal transduction pattern diagram. Auxin-inducible genes have AuxRE in their promoters, which are bound by dimers of the ARF protein family. Gene expression is prevented by recruitment of Aux/IAA transcriptional repressors to these promoters via their interaction with the ARFs. Solid arrows: verified pathway; dashed arrows: predicted pathways. DEGs encoding TIR1, ARF, and AUX/IAA are listed.

**Figure 5 ijms-25-03425-f005:**
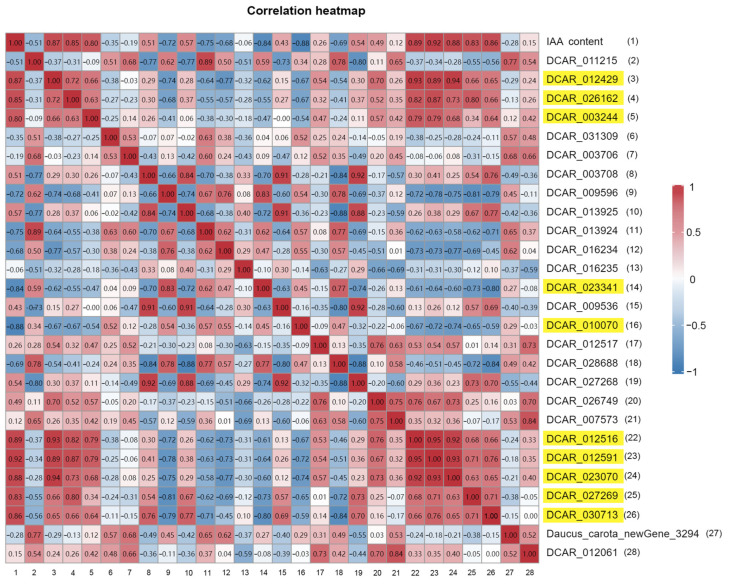
Correlation heatmap of IAA content and FPKM values of all DEGs related to IAA metabolism and signal transduction. Values are Pearson correlation coefficients; |r| ≥ 0.8 indicates gene expression significantly correlated with IAA content; yellow indicates genes significantly correlated with changes in IAA content.

**Figure 6 ijms-25-03425-f006:**
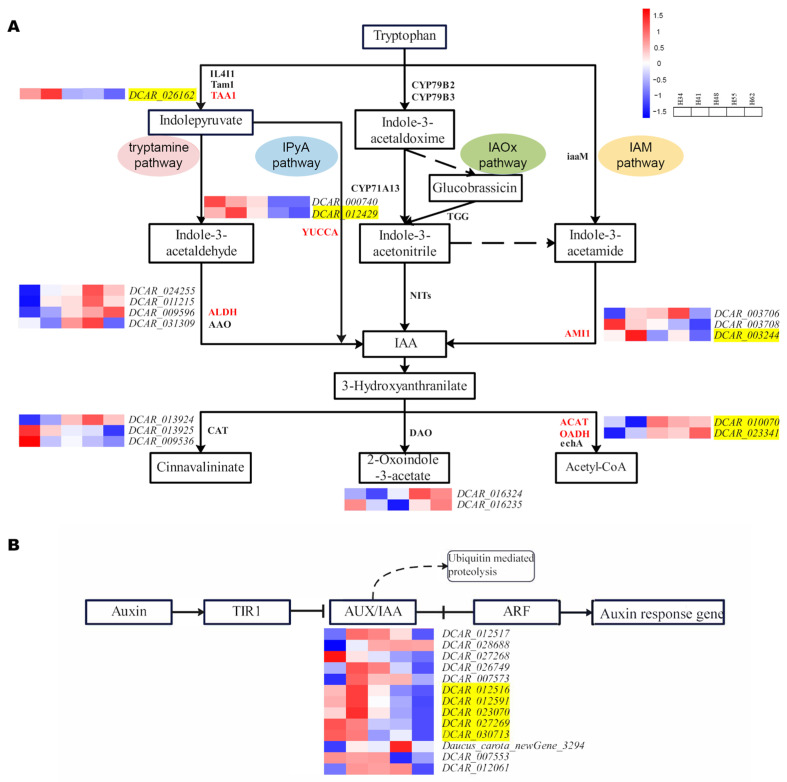
Expression analysis of DEGs during auxin metabolism. (**A**) IAA synthesis and degradation pathway; (**B**) auxin signal transduction pathway. The yellow highlights represent the genes whose expression levels are closely related to the change in IAA content, namely candidate genes.

**Table 1 ijms-25-03425-t001:** Expression of key enzyme genes in IAA synthesis process.

Enzyme Name in *Arabidopsis* or Tomato	Gene Name	FPKM					Expression Pattern		
		H34	H41	H48	H55	H62	H34 vs. H41	H41 vs. H48	H48 vs. H55
TAA1, Tryptophan aminotransferase of *Arabidopsis* 1	*DCAR_024445*	0.57 ± 0.17	0.30 ± 0.05	0.18 ± 0.06	0.18 ± 0.02	0.02 ± 0.02	down	down	-
	* **DCAR_026162** *	**9.38 ± 0.61**	**11.71 ± 0.21**	**6.21 ± 0.73**	**6.3 ± 0.63**	**5.24 ± 0.14**	up	down	up
	*DCAR_007465*	0	0	0	0	0	-	-	-
ALDH, aldehyde dehydrogenase	*Daucus_carota_newGene_5591*	0.03 ± 0.04	0.07 ± 0.02	0.11 ± 0.03	0.03 ± 0.04	0	up	up	down
	* **DCAR_011215** *	**129.65 ± 7.42**	**228.23 ± 11.79**	**237.66 ± 4.65**	**309.65 ± 4.25**	**237.22 ± 19.85**	up	up	up
	*DCAR_017329*	0.90 ± 0.22	0.66 ± 0.27	0.44 ± 0.24	0.37 ± 0.01	0.20 ± 0.01	down	down	down
	*DCAR_002508*	22.16 ± 0.23	19.18 ± 1.61	18.87 ± 0.22	18.88 ± 0.92	20.92 ± 0.61	down	down	-
	* **DCAR_024255** *	**14.33 ± 4.36**	**25.16 ± 4.59**	**27.21 ± 1.59**	**40.93 ± 4.56**	**30.50 ± 0.18**	up	up	up
	*DCAR_028207*	36.17 ± 4.82	28.67 ± 3.60	31.84 ± 2.40	20.96 ± 3.04	22.62 ± 1.08	down	up	down
	*DCAR_003558*	1.34 ± 0.61	0.67 ± 0.37	1.09 ± 0.15	0.32 ± 0.11 a	0.21 ± 0.23	down	up	down
	*DCAR_031544*	0.01 ± 0.02	0.18 ± 0.09	0.27 ± 0.02	0	0.06 ± 0.01	up	up	down
	* **DCAR_031309** *	**46.14 ± 5.28**	**38.17 ± 3.09**	**56.15 ± 3.77**	**64.09 ± 2.39**	**36.8 ± 2.43**	down	up	up
	*DCAR_007094*	0	0	0	0	0	-	-	-
	* **DCAR_009596** *	**4.50 ± 0.60**	**6.59 ± 0.53**	**10.07 ± 1.19**	**12.44 ± 2.44**	**16.13 ± 2.01**	up	up	up
	*DCAR_022687*	48.65 ± 0.68	66.18 ± 5.30	94.42 ± 3.26	114.16 ± 3.77	138.79 ± 0.89	up	up	up
YUC, YUCCA	*DCAR_011923*	0.52 ± 0.38	0.08 ± 0.06	0.07 ± 0.09	0	0	down	-	-
	* **DCAR_012429** *	**6.94 ± 0.68**	**11.19 ± 1.93**	**5.57 ± 0.63**	**3.01 ± 0.24**	**2.49 ± 0.07**	up	down	down
	*DCAR_012617*	0.28 ± 0.23	0.09 ± 0.01	0.04 ± 0.05	0.06 ± 0.02	0.09 ± 0.03	down	-	-
	*DCAR_017248*	0	0.05 ± 0.07	0	0	0.03 ± 0.04	-	-	-
	*DCAR_018364*	0.04 ± 0.05	0	0.06 ± 0.08	0.06 ± 0.02	0.02 ± 0.02	-	-	-
	*DCAR_018533*	0.43 ± 0.10	0.41 ± 0.05	0.58 ± 0.03	0.73 ± 0.22	0.56 ± 0.03	down	up	up
	* **DCAR_000740** *	**2.05 ± 0.47**	**1.14 ± 0.22**	**0.82 ± 0.10**	**0.27 ± 0.02**	**0.27 ± 0.14**	down	down	down
	*DCAR_024772*	0.33 ± 0.07	0.08 ± 0.01	0.06 ± 0.03	0.03 ± 0.02	0.02 ± 0.01	down	-	-
	*DCAR_028710*	0	0	0	0	0	-	-	-
	*DCAR_027361*	0.36 ± 0.05	0.15 ± 0.07	0.07 ± 0.01	0.02 ± 0.03	0.06 ± 0.09	down	down	-
	*DCAR_030776*	0.08 ± 0.01	0	0.01 ± 0.02	0	0.04 ± 0.06	-	-	-
	*DCAR_004697*	0.25 ± 0.06	0.02 ± 0.03	0	0	0	down	-	-
	*DCAR_000550*	0.73 ± 0.18	0.33 ± 0.10	0.36 ± 0.01	0.43 ± 0.01	0.15 ± 0.04	down	up	up
	*DCAR_000560*	0	0	0	0	0	-	-	-
	*DCAR_000561*	0	0	0	0	0	-	-	-
	*DCAR_000562*	0	0	0	0	0	-	-	-
	*DCAR_000563*	0	0	0	0	0	-	-	-
	*Daucus_carota_newGene_4145*	0	0.16 ± 0.04	0.12 ± 0.04	0	0.12 ± 0.17	up	down	down
	*Daucus_carota_newGene_6603*	0.27 ± 0.14	0.02 ± 0.03	0.07 ± 0.05	0.14 ± 0.14	0.47 ± 0.01	down	up	up
AMI1, AMIDASE1	*DCAR_017063*	7.90 ± 0.55	10.35 ± 0.16	15.21 ± 0.05	16.84 ± 1.84	12.82 ± 0.79	up	up	up
	* **DCAR_003244** *	**0.62 ± 0.02**	**1.19 ± 0.24**	**0.43 ± 0.04**	**0.63 ± 0.07**	**0.37 ± 0.01**	up	down	up
	*DCAR_029426*	0.02 ± 0.02	0.07 ± 0.01	0.08 ± 0.11	0.21 ± 0.03	0.23 ± 0.08	-	-	up
	* **DCAR_003706** *	**10.72 ± 1.77**	**23.24 ± 2.55**	**24.25 ± 1.60**	**36.87 ± 2.42**	**14.22 ± 0.95**	up	up	up
	* **DCAR_003708** *	**5.23 ± 1.36**	**1.23 ± 0.12**	**0.88 ± 0.07**	**0.41 ± 0.05**	**0.14 ± 0.04**	down	down	down
	*DCAR_005955*	0.31 ± 0.07	0.03 ± 0.03	0	0	0	down	-	-

Note: -, uniformity. Bolded genes are DEGs (FDR < 0.01, FC ≥ 1.5).

**Table 2 ijms-25-03425-t002:** Expression of key enzyme genes in IAA degradation process.

Enzyme Name in *Arabidopsis* or Tomato	Gene Name	FPKM					Expression Pattern		
		H34	H41	H48	H55	H62	H34 vs. H41	H41 vs. H48	H48 vs. H55
CAT, catalase	* **DCAR_013925** *	**128.47 ± 8.50**	**35.08 ± 8.42**	**19.34 ± 1.74**	**17.40 ± 0.86**	**4.25 ± 0.72**	down	down	down
	* **DCAR_013924** *	**586.51 ± 13.64**	**870.40 ± 72.74**	**1536.23 ± 37.53**	**2227.84 ± 103.14**	**1510.85 ± 10.22**	up	up	up
	* **DCAR_009536** *	**27.43 ± 0.60**	**11.89 ± 0.86**	**13.53 ± 0.49**	**11.61 ± 1.37**	**10.16 ± 0.17**	down	up	down
DAO, dioxygenase for auxin oxidation	*DCAR_014058*	0.05 ± 0.05	0.02 ± 0.03	0	0	0.04 ± 0.05	-	-	-
	* **DCAR_016234** *	**28.32 ± 7.51**	**17.44 ± 1.57**	**40.09 ± 0.78**	**101.25 ± 11.71**	**76.3 ± 1.65**	down	up	up
	* **DCAR_016235** *	**14.55 ± 4.27**	**3.62 ± 0.32**	**0.73 ± 0.11**	**6.83 ± 0.50**	**15.38 ± 0.17**	down	down	up
	*DCAR_002760*	1.55 ± 0.15	1.15 ± 0.20	1.24 ± 0.44	0.80 ± 0.13	0.68 ± 0.10	down	up	down
	*DCAR_006176*	6.68 ± 0.97	5.94 ± 0.78	8.24 ± 0.74	7.32 ± 0.90	6.57 ± 0.73	down	up	down
OADH, 2-oxoglutarate dehydrogenase	*DCAR_024930*	3.39 ± 0.46	2.55 ± 0.51	2.95 ± 0.05	1.79 ± 0.36	1.85 ± 0.11	down	up	down
	*DCAR_002529*	31.11 ± 3.22	27.45 ± 0.88	25.69 ± 2.58	23.18 ± 0.64	21.16 ± 1.89	down	down	down
	* **DCAR_023341** *	**5.79 ± 0.24**	**11.43 ± 1.99**	**18.48 ± 1.49**	**16.91 ± 1.05**	**26.43 ± 2.38**	up	up	down
	*DCAR_026968*	77.43 ± 11.45	64.02 ± 1.98	64.97 ± 0.65	66.75 ± 3.37	70.44 ± 0.59	down	up	down
	*DCAR_026965*	36.72 ± 1.94	40.92 ± 3.22	35.56 ± 2.07	32.28 ± 1.87	43.76 ± 1.53	up	down	down
echA, enoyl-CoA hydratase	*DCAR_030465*	59.4 ± 3.13	63.48 ± 3.43	66.35 ± 0.83	75.47 ± 0.11	64.43 ± 4.69	up	up	up
ACAT, acetyl-CoA C-acetyltransferase	*DCAR_019674*	45.67 ± 1.44	37.26 ± 3.63	43.81 ± 0.73	38.48 ± 3.11	49.68 ± 4.74	down	up	down
	* **DCAR_010070** *	**8.15 ± 0.67**	**5.76 ± 0.79**	**11.72 ± 1.00**	**10.51 ± 0.07**	**10.04 ± 0.26**	down	up	down

Note: -, uniformity. Bolded genes are DEGs (FDR < 0.01, FC ≥ 1.5).

**Table 3 ijms-25-03425-t003:** Expression of genes related to auxin signal transduction.

Enzyme Name in *Arabidopsis* or Tomato	Gene Name	FPKM					Expression Pattern		
		H34	H41	H48	H55	H62	H34 vs. H41	H41 vs. H48	H48 vs. H55
TIR1, transport inhibitor response 1	*DCAR_001970*	0.03 ± 0.06	0.32 ± 0.36	0.25 ± 0.43	0.24 ± 0.42	0.17 ± 0.29	up	down	down
	*DCAR_012874*	51.53 ± 4.03	58.83 ± 6.72	73.21 ± 8.89	71.63 ± 2.89	80.20 ± 9.75	up	up	down
	*DCAR_019294*	8.37 ± 0.84	8.00 ± 1.18	8.62 ± 0.99	11.14 ± 3.56	8.51 ± 2.09	down	up	up
	* **DCAR_028134** *	**6.37 ± 1.29**	**6.02 ± 0.90**	**3.58 ± 1.23**	**2.82 ± 0.91**	**2.20 ± 0.46**	down	down	down
	*DCAR_003381*	0	0	0.03 ± 0.05	0.02 ± 0.03	0	-	-	-
AUX/IAA, auxin/indole-3-acetic acid	* **Daucus_carota_newGene_6588** *	**2.91 ± 0.42**	**2.72 ± 0.40**	**1.79 ± 0.93**	**0.69 ± 0.28**	**0.69 ± 0.31**	down	down	down
	* **DCAR_010763** *	**288.00 ± 19.62**	**270.40 ± 31.12**	**230.32 ± 14.44**	**176.46 ± 1.30**	**185.92 ± 18.42**	down	down	down
	* **DCAR_012516** *	**18.48 ± 3.68**	**29.32 ± 5.09**	**15.11 ± 1.59**	**9.23 ± 2.73**	**7.60 ± 1.00**	up	down	down
	* **DCAR_012591** *	**50.50 ± 5.68**	**70.74 ± 11.44**	**39.19 ± 0.63**	**30.46 ± 5.95**	**22.51 ± 6.77**	up	down	down
	*DCAR_014699*	1.16 ± 0.80	0.21 ± 0.18	0.05 ± 0.08	0	0	-	-	-
	* **DCAR_018241** *	**3.03 ± 0.61**	**3.14 ± 0.13**	**1.65 ± 0.71**	**0.77 ± 0.45**	**0.83 ± 0.52**	up	down	down
	* **DCAR_018447** *	**8.65 ± 1.81**	**8.76 ± 1.69**	**4.79 ± 1.62**	**5.01 ± 1.38**	**1.86 ± 0.51**	up	down	up
	* **DCAR_002354** *	**10.73 ± 1.19**	**11.41 ± 4.10**	**7.03 ± 1.15**	**7.28 ± 1.70**	**3.72 ± 0.97**	up	down	up
	* **DCAR_023070** *	**41.69 ± 1.75**	**62.10 ± 5.52**	**39.80 ± 1.75**	**30.24 ± 6.94**	**24.43 ± 2.99**	up	down	down
	* **DCAR_024202** *	**13.01 ± 3.44**	**12.45 ± 5.81**	**9.62 ± 2.76**	**9.71 ± 3.02**	**6.78 ± 3.18**	down	down	up
	* **DCAR_026353** *	**20.73 ± 1.80**	**18.59 ± 0.56**	**14.79 ± 1.08**	**13.98 ± 1.13**	**15.45 ± 0.41**	down	down	down
	* **DCAR_027352** *	**132.74 ± 9.55**	**170.35 ± 23.25**	**121.75 ± 16.38**	**85.32 ± 12.35**	**75.53 ± 25.26**	up	down	down
	* **DCAR_027269** *	**9.41 ± 2.80**	**7.56 ± 1.89**	**3.34 ± 0.57**	**3.15 ± 1.45**	**1.92 ± 0.75**	down	down	down
	* **DCAR_030712** *	**59.43 ± 9.45**	**73.77 ± 14.10**	**41.94 ± 12.29**	**43.31 ± 8.77**	**25.12 ± 6.31**	up	down	up
	* **DCAR_030713** *	**178.48 ± 32.21**	**160.68 ± 24.05**	**78.81 ± 16.93**	**102.42 ± 19.16**	**62.01 ± 18.06**	down	down	up
ARF, auxin response factor	*DCAR_011930*	8.75 ± 0.30	9.68 ± 0.93	11.14 ± 0.83	12.17 ± 0.62	11.68 ± 0.48	up	up	up
	* **DCAR_015554** *	**13.13 ± 1.64**	**14.88 ± 1.57**	**11.97 ± 1.11**	**11.87 ± 1.24**	**11.50 ± 1.87**	up	down	down
	* **DCAR_014788** *	**1.10 ± 0.20**	**0.86 ± 0.18**	**0.51 ± 0.19**	**0.34 ± 0.10**	**0.43 ± 0.26**	down	down	down
	*DCAR_014129*	26.39 ± 2.26	38.22 ± 4.57	35.69 ± 2.22	31.88 ± 2.46	35.52 ± 2.70	up	down	down
	* **DCAR_017496** *	**32.13 ± 0.11 ab**	**37.99 ± 5.71**	**30.70 ± 1.62**	**25.91 ± 1.08**	**29.77 ± 5.90**	up	down	down
	* **DCAR_020464** *	**28.80 ± 1.51**	**29.73 ± 2.44**	**25.07 ± 5.10**	**22.47 ± 2.06**	**15.05 ± 1.77**	up	down	down
	* **DCAR_005400** *	**1.05 ± 0.19**	**0.77 ± 0.27**	**0.52 ± 0.12**	**0.56 ± 0.17**	**0.40 ± 0.15**	down	down	up
	* **DCAR_006053** *	**15.82 ± 0.32**	**26.08 ± 2.39**	**17.68 ± 0.49**	**11.17 ± 0.66**	**11.35 ± 1.22**	up	down	down
	*DCAR_006434*	4.79 ± 0.74	2.71 ± 0.46	1.37 ± 0.15	1.00 ± 0.04	0.77 ± 0.26	down	down	down

Note: -, uniformity. Bolded genes are DEGs (FDR < 0.01, FC ≥ 1.5).

**Table 4 ijms-25-03425-t004:** DEGs of tryptophan metabolism in root expansion at different stages.

Auxin Metabolism	Key Enzyme	Coding Gene Name
IAA synthesis process	ALDH	*DCAR_011215*, *DCAR_031309*, *DCAR_009596*
	YUCCA	*DCAR_012429*
	TAA1	*DCAR_026162*
	AMI1	*DCAR_003244*, *DCAR_003706*, *DCAR_003708*
IAA catabolic process	CAT	*DCAR_013925*, *DCAR_013924*, *DCAR_009536*
	DAO	*DCAR_016234*, *DCAR_016235*
	OADH	*DCAR_023341*
	ACAT	*DCAR_010070*
Auxin signal transduction pathway	AUX/IAA	*DCAR_012517*, *DCAR_028688*, *DCAR_027268*, *DCAR_026749*, *DCAR_007573*, *DCAR_012516*, *DCAR_012591*, *DCAR_023070*, *DCAR_027269*, *DCAR_030713*, *Daucus_carota_newGene_3294*, *DCAR_012061*

**Table 5 ijms-25-03425-t005:** DEGs of root expansion at different stages.

Stage	Auxin Metabolism	Gene Name	Homologous Gene in *Arabidopsis*	Enzyme Name	Gene GO Annotation	log_2_FC
H41 vs. H48	Synthesis process	*DCAR_026162*	*TAR2*	TAR, L-tryptophan-pyruvate aminotransferase	Carbon-sulfur lyase activity (GO:0016846)	−0.78
		*DCAR_012429*	*YUCCA10*	YUCCA, indole-3-pyruvate monooxygenase	N,N-dimethylaniline monooxygenase activity (GO:0004499); NADP binding (GO:0050661)	−0.86
		*DCAR_003244*	*AMI1*	AMI1, amidase	-	−1.10
	Catabolism process	*DCAR_010070*	*AACT1*	ACAT, acetyl-CoA C-acetyltransferase	Transferase activity (GO:0016747)	0.85
	Auxin signal transduction pathway	*DCAR_012516*	*IAA22*	auxin-responsive protein IAA	regulation of transcription, DNA-templated (GO:0006355); auxin-activated signaling pathway (GO:0009734)	−0.85
		*DCAR_012591*	*IAA13*	auxin-responsive protein IAA	regulation of transcription, DNA-templated (GO:0006355); auxin-activated signaling pathway (GO:0009734)	−0.80
		*DCAR_023070*	*IAA27*	auxin-responsive protein IAA	regulation of transcription, DNA-templated (GO:0006355); auxin-activated signaling pathway (GO:0009734)	−0.60
		*DCAR_027269*	*IAA14*	auxin-responsive protein IAA	regulation of transcription, DNA-templated (GO:0006355); auxin-activated signaling pathway (GO:0009734)	−0.93
		*DCAR_030713*	*IAA7*	auxin-responsive protein IAA	regulation of transcription, DNA-templated (GO:0006355); auxin-activated signaling pathway (GO:0009734)	−0.92
H55 vs. H62	Catabolism process	*DCAR_023341*	*LPD1*	OADH, 2-oxoglutarate dehydrogenase	Dihydrolipoyl dehydrogenase activity (GO:0004148); flavin adenine dinucleotide binding (GO:0050660)	0.65

## Data Availability

Publicly available datasets were analyzed in this study. These data can be found here: https://www.ncbi.nlm.nih.gov/bioproject/PRJNA821411 (accessed on 10 April 2022).

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
