# Peer review of "Transcriptome Sequencing Reveals the Mechanism of Auxin Regulation during Root Expansion in Carrot"

_ijms, 2024, doi:10.3390/ijms25063425_

Round 1
Reviewer 1 Report (Previous Reviewer 2)
Comments and Suggestions for Authors
Review ijms 2894286
In the updated version of the manuscript, the authors changed the title of the chapter4.5. Screening of Root Expansion-Related Genes to IAA Metabolism-Related Genes during Root Expansion, but still did not provide data on the relationship between auxin genes and genes regulating cell expansion, promising to do so in the next study. I can only repeat once again that that without these data it is impossible to claim to identify the mechanisms “of Auxin Regulation of Root Expansion in Carrot”. In this regard, I would recommend that the authors at least change the title of the article to “Transcriptome Sequencing Reveals the Mechanism of Auxin Regulation during Root Expansion in Carrot” which would more accurately reflect the essence of the experiments carried out by the authors.
t
Author Response
Response to Reviewer 1 Comments
|
||
1. Summary |
|
|
We sincerely thank you for taking your valuable time and energy to review this manuscript. According to your professional and constructive comments, we have revised the manuscript point-to-point in accordance with your valuable suggestions, and the corresponding corrections were highlighted. We asked Prof. Zhiqian Liu in the Agriculture Victoria of the Centre for AgriBiosciences to help polishing the language of the manuscript and the polished parts were marked as red in the “Revised manuscript (with highlighted marks) ijms-2894286”. The revised information and responses are as follows. |
||
2. Questions for General Evaluation |
Reviewer’s Evaluation |
Response and Revisions |
Does the introduction provide sufficient background and include all relevant references? |
Can be improved |
|
Are all the cited references relevant to the research? |
Can be improved |
|
Is the research design appropriate? |
Can be improved |
|
Are the methods adequately described? |
Must be improved |
|
Are the results clearly presented? |
Must be improved |
|
Are the conclusions supported by the results? |
Not applicable |
We have changed the title of the article according to the reviewer’s comment. Therefore, the results can support the conclusion. |
3. Point-by-point response to Comments and Suggestions for Authors |
||
Comments 1: In the updated version of the manuscript, the authors changed the title of the chapter4.5. Screening of Root Expansion-Related Genes to IAA Metabolism-Related Genes during Root Expansion, but still did not provide data on the relationship between auxin genes and genes regulating cell expansion, promising to do so in the next study. I can only repeat once again that that without these data it is impossible to claim to identify the mechanisms “of Auxin Regulation of Root Expansion in Carrot”. In this regard, I would recommend that the authors at least change the title of the article to “Transcriptome Sequencing Reveals the Mechanism of Auxin Regulation during Root Expansion in Carrot” which would more accurately reflect the essence of the experiments carried out by the authors. |
||
Response 1: Thank you for your valuable suggestion. We have changed the title of the article to “Transcriptome Sequencing Reveals the Mechanism of Auxin Regulation during Root Expansion in Carrot” in line 3 and highlighted them. Please check it. |
||
4. Response to Comments on the Quality of English Language |
||
Point 1: English language fine. No issue detected. |
||
Response 1: Thank you for your patient review. |
||
5. Additional clarifications |
||
We have nothing else to elaborate on and thank you again for your valuable comments. |
Reviewer 2 Report (Previous Reviewer 1)
Comments and Suggestions for Authors
The manuscript entitled “Transcriptome Sequencing Reveals the Mechanism of Auxin Regulation of Root Expansion in Carrot” by Li and collaborators presents present a transcriptomic and metabolomic analysis of the agricultural species Daucus carota. The approach and the analyses present in the MS are of interest, as well as the intention to describe the role of a very relevant hormone in the root such as auxins. However, the MS needs, from this reviewer's point of view, a restructuring of the material presented in order to better understand the results presented.
The authors should take the following major points into account
1. Tables 2 and 3 should be summarised. Only the DEG genes should be shown, as well as their annotation of the most likely orthologue extracted from supplementary table 2.
2.- In order to better explore the transcriptomic data, a supplementary table with the expression values of at least the genes treated in the MS should be included. This information is not useful in tables 2 and 3 of the main body.
3.- In the supplementary table S2, where did you get the annotation information from MapMan? From plaza dicots? You should elaborate more on this information and include the identity/homology/ortology values obtained from your gene annotation analysis.
4.- About the "Sample Repeatability Test" section (lines 101-108), the authors should include more analysis to support this stament. For example a heat map with Hierarchical clustering to see how all the samples cluster.
5.- From this reviewer's point of view, the system used by the authors does not reflect the full dynamics of auxins in the carrot root. There are studies such as Perrin et al. 2017, in which they find differences between different tissues in carrot root. The authors should take this study into account when discussing their results.
Perrin F, Hartmann L, Dubois-Laurent C, Welsch R, Huet S, Hamama L, Briard M, Peltier D, Gagné S, Geoffriau E. Carotenoid gene expression explains the difference of carotenoid accumulation in carrot root tissues. Planta. 2017 Apr;245(4):737-747. doi: 10.1007/s00425-016-2637-9. Epub 2016 Dec 20. PMID: 27999990.
Figure 5 must be an improvement, it is impossible to see the numbers. They should colour the correlation values. It is necessary to detail how the analysis has been done. Have all auxin-related genes been used? Only those of sistensis?
Minor changes:
Figure 1 should have a white background, like the rest of the figures in the MS.
Authors must indicate the expression cut-off used for their RNA-Seq analysis.
Author Response
Response to Reviewer 2 Comments
|
||
1. Summary |
|
|
We sincerely thank you for taking your valuable time and energy to review this manuscript. According to your professional and constructive comments, we have revised the manuscript point-to-point in accordance with your valuable suggestions, and the corresponding corrections were highlighted. We asked Prof. Zhiqian Liu in the Agriculture Victoria of the Centre for AgriBiosciences to help polishing the language of the manuscript and the polished parts were marked as red in the “Revised manuscript (with highlighted marks) ijms-2894286”. The revised information and responses are as follows. |
||
2. Questions for General Evaluation |
Reviewer’s Evaluation |
Response and Revisions |
Does the introduction provide sufficient background and include all relevant references? |
Can be improved |
|
Are all the cited references relevant to the research? |
Can be improved |
|
Is the research design appropriate? |
Can be improved |
|
Are the methods adequately described? |
Can be improved |
|
Are the results clearly presented? |
Can be improved |
|
Are the conclusions supported by the results? |
Must be improved |
|
3. Point-by-point response to Comments and Suggestions for Authors |
||
Comments 1: The manuscript entitled “Transcriptome Sequencing Reveals the Mechanism of Auxin Regulation of Root Expansion in Carrot” by Li and collaborators presents present a transcriptomic and metabolomic analysis of the agricultural species Daucus carota. The approach and the analyses present in the MS are of interest, as well as the intention to describe the role of a very relevant hormone in the root such as auxins. However, the MS needs, from this reviewer's point of view, a restructuring of the material presented in order to better understand the results presented. The authors should take the following major points into account 1. Tables 2 and 3 should be summarised. Only the DEG genes should be shown, as well as their annotation of the most likely orthologue extracted from supplementary table 2. |
||
Response 1: Thank you for your valuable comments. We have removed some genes and kept DEGs and some genes with annotation of the most likely orthologue genes in the degradation and signaling transduction in Tables 2 and 3 according to your comment and marked them highlighted. |
||
Comments 2: 2.- In order to better explore the transcriptomic data, a supplementary table with the expression values of at least the genes treated in the MS should be included. This information is not useful in tables 2 and 3 of the main body. |
||
Response 2: In order to understand the gene expression level more intuitively, we hope to retain the FPKM values in Tables 2 and 3 like Table 1. Additionally, we have removed the FPKM values and annotations of all genes encoding auxin metabolism and signal transduction in supplementary Table S2 and highlighted them. |
||
Comments 3: 3.- In the supplementary table S2, where did you get the annotation information from MapMan? From plaza dicots? You should elaborate more on this information and include the identity/homology/ortology values obtained from your gene annotation analysis. |
||
Response 3: We obtained the gene annotation information from Dicots PLAZA 5.0 (https://bioinformatics.psb.ugent.be/plaza/versions/plaza_v5_dicots/). We have added the detailed information in lines 413-416 of 4.3.2 section and highlighted them. We have added the homology/orthology values in Table S2 and highlighted them. Please check it. |
||
Comments 4: 4.- About the "Sample Repeatability Test" section (lines 101-108), the authors should include more analysis to support this stament. For example a heat map with Hierarchical clustering to see how all the samples cluster. |
||
Response 4: We have added the heat map with hierarchical clustering in supplementary materials and named it Figure S2. Please check it. |
||
Comments 5: 5.- From this reviewer's point of view, the system used by the authors does not reflect the full dynamics of auxins in the carrot root. There are studies such as Perrin et al. 2017, in which they find differences between different tissues in carrot root. The authors should take this study into account when discussing their results. Perrin F, Hartmann L, Dubois-Laurent C, Welsch R, Huet S, Hamama L, Briard M, Peltier D, Gagné S, Geoffriau E. Carotenoid gene expression explains the difference of carotenoid accumulation in carrot root tissues. Planta. 2017 Apr;245(4):737-747. doi: 10.1007/s00425-016-2637-9. Epub 2016 Dec 20. PMID: 27999990. |
||
Response 5: Thank you for your valuable suggestion. We have added this reference in lines 319-323 in “3. Discussion” section and highlighted them. |
||
Comments 6: Figure 5 must be an improvement, it is impossible to see the numbers. They should colour the correlation values. It is necessary to detail how the analysis has been done. Have all auxin-related genes been used? Only those of sistensis? |
||
Response 6: We have changed Figure 5 as the correlation heatmap with Pearson correlation values and uploaded Figure 5 with 300 dpi separately. In addition, we also added the method in 4.6. of Materials and Methods section and highlighted them. The correlation analysis between the IAA content and the FPKM values were performed for all of DEGs related to IAA metabolism and signal transduction (Table 4). |
||
Comments 7: Figure 1 should have a white background, like the rest of the figures in the MS. |
||
Response 7: We have modified the background of Figure 1 to white in accordance with your recommendation. Please check it. |
||
Comments 8: Authors must indicate the expression cut-off used for their RNA-Seq analysis. |
||
Response 8: We have added the cut-off E-value in line 416 and highlighted them. |
||
4. Response to Comments on the Quality of English Language |
||
Point 1: I am not qualified to assess the quality of English in this paper. |
||
Response 1: Thank you for your patient review. |
||
5. Additional clarifications |
||
We have nothing else to elaborate on and thank you again for your valuable comments. |
Reviewer 3 Report (New Reviewer)
Comments and Suggestions for Authors
- On Page 1, Line 43: The statement "Auxin, the first plant hormone to be discovered," is inaccurate. Auxin is not a single hormone but rather a class of hormones.
- On Page 2, in section 2.1 and Figure 1: Figure 1A to 1D were not presented in the correct order. It is recommended to rearrange Figure 1A to 1D according to the text.
- The quality of Figure 1A and 1B needs improvement.
- On Page 4, Lines 119-134: The sentences do not pertain to the results of the study and should be considered for inclusion in the introduction or discussion sections.
- Please include details about software and statistical analyses in the "materials and methods" section.
Comments on the Quality of English LanguageDear Editor,
Thank you for the invitation. The manuscript is and designed properly. However, some minor revisions are needed before publishing. in addition, the authors did not used 2024 and 2023 references in the manuscript.
Best regards,
Jahad Soorni
Author Response
Response to Reviewer 3 Comments
|
||
1. Summary |
|
|
We sincerely thank you for taking your valuable time and energy to review this manuscript. According to your professional and constructive comments, we have revised the manuscript point-to-point in accordance with your valuable suggestions, and the corresponding corrections were highlighted. We asked Prof. Zhiqian Liu in the Agriculture Victoria of the Centre for AgriBiosciences to help polishing the language of the manuscript and the polished parts were marked as red in the “Revised manuscript (with highlighted marks) ijms-2894286”. The revised information and responses are as follows. |
||
2. Questions for General Evaluation |
Reviewer’s Evaluation |
Response and Revisions |
Does the introduction provide sufficient background and include all relevant references? |
Can be improved |
|
Are all the cited references relevant to the research? |
Yes |
|
Is the research design appropriate? |
Yes |
|
Are the methods adequately described? |
Can be improved |
|
Are the results clearly presented? |
Yes |
|
Are the conclusions supported by the results? |
Yes |
|
3. Point-by-point response to Comments and Suggestions for Authors |
||
Comments 1: - On Page 1, Line 43: The statement "Auxin, the first plant hormone to be discovered," is inaccurate. Auxin is not a single hormone but rather a class of hormones. |
||
Response 1: Thank you for your comment. We have changed the description in line 43 and highlighted them. |
||
Comments 2: - On Page 2, in section 2.1 and Figure 1: Figure 1A to 1D were not presented in the correct order. It is recommended to rearrange Figure 1A to 1D according to the text. |
||
Response 2: We have changed their order in Figure 1 according to your recommendation. Please check it. |
||
Comments 3: - The quality of Figure 1A and 1B needs improvement. |
||
Response 3: We have improved the quality of Figures 1A and 1B (now they are named Figures 1C and 1D) and uploaded Figure 1 with 300 dpi separately. |
||
Comments 4: - On Page 4, Lines 119-134: The sentences do not pertain to the results of the study and should be considered for inclusion in the introduction or discussion sections. |
||
Response 4: We think the content of the description of auxin metabolism is not necessary in the introduction, however, if not stated in the results and analysis, it would affect the reader's understanding of the content, so we hope to retain it there. |
||
Comments 5: - Please include details about software and statistical analyses in the "materials and methods" section. |
||
Response 5: Thank you for your recommendation. We have added the details about software and statistical analyses in lines 457-462 of the “materials and methods” section and highlighted them. |
||
4. Response to Comments on the Quality of English Language |
||
Point 1: Minor editing of English language required. |
||
Response 1: We asked Prof. Zhiqian Liu in the Agriculture Victoria of the Centre for AgriBiosciences to help polishing the manuscript and marked the modifications as red. Thank you for your valuable comments. |
||
Point 2: Dear Editor, Thank you for the invitation. The manuscript is and designed properly. However, some minor revisions are needed before publishing. in addition, the authors did not used 2024 and 2023 references in the manuscript. Best regards, Jahad Soorni |
||
Response 2: Thank you for your patient review. We have added some references published in 2023 and 2024 in lines 49-51, 55-57, and 354-356 and highlighted them. |
||
5. Additional clarifications |
||
We have nothing else to elaborate on and thank you again for your valuable comments. |
Reviewer 4 Report (New Reviewer)
Comments and Suggestions for Authors
This paper conducted comprehensive RNA-seq analysis on root development of carrot, attempting to study the mechanism of auxin regulation of root expansion. However, partial data are too rough and of low quality, or lacks essential details. It is insufficient to support the results claimed by the authors.
Figures are relatively independent parts; therefore, it is necessary to provide details to help readers understand. However, Figure 4 is missing a figure legend. Similarly, many other figures (e.g. Figure S2, Figure 1) lack necessary captions, too.
Figure S2 (and the line chart in Figure 1) lacks significance test, so they cannot explain any issues.
Some figures are very unclear and the text in the pictures cannot be read. e.g. Figure 1 (especially A and B), Figure 3(IAA metabolism process and key enzymes), Figure 5, and Figure 6.
The authors only provide the expression levels of a small number of analyzed genes (IAA-related genes) in the main text, but do not provide the raw data of transcriptome sequencing (expression levels of different genes in different samples). Representative data should be provided in attachments for reference and replication by others.
Author Response
Response to Reviewer 4 Comments
|
||
1. Summary |
|
|
We sincerely thank you for taking your valuable time and energy to review this manuscript. According to your professional and constructive comments, we have revised the manuscript point-to-point in accordance with your valuable suggestions and the corresponding corrections were highlighted. We asked Prof. Zhiqian Liu in the Agriculture Victoria of the Centre for AgriBiosciences to help polishing the language of the manuscript and the polished parts were marked as red in the “Revised manuscript (with highlighted marks) ijms-2894286”. The revised information and responses are as follows. |
||
2. Questions for General Evaluation |
Reviewer’s Evaluation |
Response and Revisions |
Does the introduction provide sufficient background and include all relevant references? |
Yes |
|
Are all the cited references relevant to the research? |
Yes |
|
Is the research design appropriate? |
Can be improved |
|
Are the methods adequately described? |
Can be improved |
|
Are the results clearly presented? |
Must be improved |
|
Are the conclusions supported by the results? |
Must be improved |
|
3. Point-by-point response to Comments and Suggestions for Authors |
||
Comments 1: This paper conducted comprehensive RNA-seq analysis on root development of carrot, attempting to study the mechanism of auxin regulation of root expansion. However, partial data are too rough and of low quality, or lacks essential details. It is insufficient to support the results claimed by the authors. Figures are relatively independent parts; therefore, it is necessary to provide details to help readers understand. However, Figure 4 is missing a figure legend. Similarly, many other figures (e.g. Figure S2, Figure 1) lack necessary captions, too. |
||
Response 1: Thank you for your comments. We have added figure legends in Figure 1, Figure 4, and Figure S2 (now it was named Figure S3) and highlighted them. |
||
Comments 2: Figure S2 (and the line chart in Figure 1) lacks significance test, so they cannot explain any issues. |
||
Response 2: We have added the significance in Figure 1 and Figure S2 (now it was named Figure S3). Please check it. |
||
Comments 3: Some figures are very unclear and the text in the pictures cannot be read. e.g. Figure 1 (especially A and B), Figure 3(IAA metabolism process and key enzymes), Figure 5, and Figure 6. |
||
Response 3: We have replaced the original with more clear figures in the manuscript and uploaded Figure 1, Figure 3, Figure 5, and Figure 6 with 300 dpi separately. |
||
Comments 4: The authors only provide the expression levels of a small number of analyzed genes (IAA-related genes) in the main text, but do not provide the raw data of transcriptome sequencing (expression levels of different genes in different samples). Representative data should be provided in attachments for reference and replication by others. |
||
Response 4: We have deposited the RNA-seq data into the NCBI SRA database under accession number PRJNA821411 (https://www.ncbi.nlm.nih.gov/sra/PRJNA821411) and corresponding sequence data are referenced to BioSample accessions SAMN27097857 (for H34), SAMN27097858 (for H41), SAMN27097859 (for H48), SAMN27097860 (for H55), and SAMN27097861 (for H62). |
||
4. Response to Comments on the Quality of English Language |
||
Point 1: I am not qualified to assess the quality of English in this paper. |
||
Response 1: Thank you for your patient review. |
||
5. Additional clarifications |
||
We have nothing else to elaborate on and thank you again for your valuable comments. |
Round 2
Reviewer 1 Report (Previous Reviewer 2)
Comments and Suggestions for Authors
Review ijms 2894286
The authors introduced a modified version of the title of the manuscript, which more closely reflects the essence of the data they obtained, and polished the text. In corrected form, the manuscript can be accepted for publication.
Author Response
Response to Reviewer 1 Comments
|
||
1. Summary |
|
|
We appreciate you for taking your valuable time and effort to review this manuscript, and thank you for your positive comment. |
||
2. Questions for General Evaluation |
Reviewer’s Evaluation |
Response and Revisions |
Does the introduction provide sufficient background and include all relevant references? |
Yes |
|
Are all the cited references relevant to the research? |
Yes |
|
Is the research design appropriate? |
Yes |
|
Are the methods adequately described? |
Yes |
|
Are the results clearly presented? |
Yes |
|
Are the conclusions supported by the results? |
Yes |
|
3. Point-by-point response to Comments and Suggestions for Authors |
||
Comments 1: The authors introduced a modified version of the title of the manuscript, which more closely reflects the essence of the data they obtained, and polished the text. In corrected form, the manuscript can be accepted for publication. |
||
Response 1: We are grateful for your patient review and thank you for your decision on this manuscript. |
||
4. Response to Comments on the Quality of English Language |
||
Point 1: English language fine. No issue detected. |
||
Response 1: Thank you for your patient review. |
||
5. Additional clarifications |
||
We have nothing else to elaborate on and thank you again for your valuable comments. |
Reviewer 2 Report (Previous Reviewer 1)
Comments and Suggestions for Authors
The new version of manuscript entitled “Transcriptome Sequencing Reveals the Mechanism of Auxin Regulation of Root Expansion in Carrot” by Li and collaborators has been improved as the authors have included the comments of the different reviewers. However, this reviewer has not located one of the authors' responses, and some changes are needed.
Line 416 does not contain the cut-off indicated by the authors. It is necessary for the authors to indicate what is the minimum value of FPKMs they have used to determine that a gene is expressed or not in their experiment.
In addition to depositing the data from their experiment at NCBI, this reviewer would expect a table with the expression of all genes, as well as their basic annotation, to understand what role auxin-related genes play within the context of the root transcriptome. This would be very positive for the article.
In tables 1,2 and 3 change the arrows to the words up and down regulated.
In figure 4 the gene expression values should be included, perhaps more visually than in a table, which can be made supplementary.
Author Response
Response to Reviewer 2 Comments
|
||
1. Summary |
|
|
We sincerely thank you for taking your valuable time and energy to review this manuscript. According to your professional and constructive comments, we have revised the manuscript point-to-point in accordance with your valuable suggestions, and the corresponding corrections were highlighted in the “Revised manuscript (with highlighted marks) ijms-2894286”. The revised information and responses are as follows. |
||
2. Questions for General Evaluation |
Reviewer’s Evaluation |
Response and Revisions |
Does the introduction provide sufficient background and include all relevant references? |
Yes |
|
Are all the cited references relevant to the research? |
Yes |
|
Is the research design appropriate? |
Can be improved |
|
Are the methods adequately described? |
Can be improved |
|
Are the results clearly presented? |
Can be improved |
|
Are the conclusions supported by the results? |
Can be improved |
|
3. Point-by-point response to Comments and Suggestions for Authors |
||
Comments 1: The new version of manuscript entitled “Transcriptome Sequencing Reveals the Mechanism of Auxin Regulation of Root Expansion in Carrot” by Li and collaborators has been improved as the authors have included the comments of the different reviewers. However, this reviewer has not located one of the authors' responses, and some changes are needed.
Line 416 does not contain the cut-off indicated by the authors. It is necessary for the authors to indicate what is the minimum value of FPKMs they have used to determine that a gene is expressed or not in their experiment. |
||
Response 1: Thank you for your valuable comment. In RNA-seq, all genes whose FPKM cut-off is greater than zero are considered. When analyzing gene expression, we only consider the differences in expression in different periods, and set FDR < 0.01 and FC ≥ 1.5 as thresholds for screening. |
||
Comments 2: In addition to depositing the data from their experiment at NCBI, this reviewer would expect a table with the expression of all genes, as well as their basic annotation, to understand what role auxin-related genes play within the context of the root transcriptome. This would be very positive for the article. |
||
Response 2: Because the original data file is too large, we think it is not suitable to be attached as supplementary material to the article, and the disclosure of these data will affect our subsequent analysis in other aspects. Hope you understand. |
||
Comments 3: In tables 1,2 and 3 change the arrows to the words up and down regulated. |
||
Response 3: Thank you for your valuable suggestion. We have changed the arrows to the words up and down regulated in Tables 1, 2, and 3 and highlighted them. Please check it. |
||
Comments 4: In figure 4 the gene expression values should be included, perhaps more visually than in a table, which can be made supplementary. |
||
Response 4: Thank you for your suggestion. The genes in Figure 4 are auxin signaling transduction-related DEGs and their gene expression values and annotations are listed in Table 3 and Table S2. We added the gene expression values in Figure 4 by heatmap. Please check it. |
||
4. Response to Comments on the Quality of English Language |
||
Point 1: I am not qualified to assess the quality of English in this paper |
||
Response 1: Thank you for your patient review. |
||
5. Additional clarifications |
||
We have nothing else to elaborate on and thank you again for your valuable comments. |
Reviewer 4 Report (New Reviewer)
Comments and Suggestions for Authors
Figure S3 is missing error bars and needs to be added based on calculations from the original data.
Author Response
Response to Reviewer 4 Comments
|
||
1. Summary |
|
|
We sincerely thank you for taking your valuable time and energy to review this manuscript. According to your professional and constructive comments, we have revised the manuscript point-to-point in accordance with your valuable suggestions, and the corresponding corrections were highlighted in the “Revised manuscript (with highlighted marks) ijms-2894286”. The revised information and responses are as follows. |
||
2. Questions for General Evaluation |
Reviewer’s Evaluation |
Response and Revisions |
Does the introduction provide sufficient background and include all relevant references? |
Yes |
|
Are all the cited references relevant to the research? |
Yes |
|
Is the research design appropriate? |
Yes |
|
Are the methods adequately described? |
Yes |
|
Are the results clearly presented? |
Can be improved |
|
Are the conclusions supported by the results? |
Yes |
|
3. Point-by-point response to Comments and Suggestions for Authors |
||
Comments 1: Figure S3 is missing error bars and needs to be added based on calculations from the original data. |
||
Response 1: Thank you for your valuable suggestion. We have added the error bars in Figure S3. Please check it. |
||
4. Response to Comments on the Quality of English Language |
||
Point 1: I am not qualified to assess the quality of English in this paper |
||
Response 1: Thank you for your patient review. |
||
5. Additional clarifications |
||
We have nothing else to elaborate on and thank you again for your valuable comments. |
This manuscript is a resubmission of an earlier submission. The following is a list of the peer review reports and author responses from that submission.
Round 1
Reviewer 1 Report
Comments and Suggestions for Authors
The manuscript entitled “Transcriptome Sequencing Reveals the Mechanism of Auxin Regulation of Root Expansion in Carrot” by Li and collaborators presents present a transcriptomic and metabolomic analysis of the agricultural species Daucus carota. The approach and the analyses present in the MS are of interest, as well as the intention to describe the role of a very relevant hormone in the root such as auxins. However, it is necessary that the authors detail more precisely the genetic annotation, since carrot is not a model species, and it is also necessary that their experimental system really reflects the auxin dynamics in an organism as complex as the carrot root.
The authors should take the following major points into account
On the experimental system. The authors only analyse the gene expression and meatabolomics of one region of the root, line 338 "middle part of carrot roots with similar growth". It is necessary for the authors to demonstrate that this region is indeed representative of the whole root. In their results and conclusions, they assume that what happens in this region can be extrapolated to the rest of the whole organ. In fact, auxins are a hormone that undergoes extensive movement throughout the plant, and in the root it has a polar transport. For this reason, it is necessary to show that this region is representative of the whole root.
As for gene annotation, when working with a non-model organism, such as Arabidopsis thaliana, it is necessary to provide a lot of evidence for gene annotation. The authors do not indicate how they have identified the genes, how they have annotated them and how they have identified the different genes associated with auxin biosynthesis, metabolism and signal transduction. This point should be well explained in the materials and methods section.
In fact, there are incorrect annotations of the genes, as an example;
DCAR_013008 annotated as TAA1 by the authors, appears as orthologue of the FLOE1 gene (AT4G28300) in Esemble database (https://plants.ensembl.org/Daucus_carota/Gene/Compara_Ortholog?db=core;g=DCAR_013008;r=4:33572028-33577084;t=KZM99630). The same result is obtained with another gene also identified as TAA1, DCAR_021243, in this case through a phylogeny also from Emsembl database (https://plants.ensembl.org/Daucus_carota/Gene/Compara_Tree?anc=4697284;db=core;g=DCAR_021243;g1=AT4G28300;r=6:24748057-24751437;t=KZM91392).
These are just two examples where the annotations or associations made by the authors do not match those present in the databases, and the same is true when checked in the Plaza Dicots database (https://bioinformatics.psb.ugent.be/plaza/versions/plaza_v5_dicots/).
Authors should indicate and correct the annotations they indicate. In the case of complex families such as ARF and YUCCA, the construction of phylogenetic trees is also necessary.
Finally, regarding the RNA-Seq analysis, the authors only indicate which software they use for mapping, in this case HISAT2 (Line 365), but they do not detail how they do the gene counting, and they do not indicate which gene structural prediction (GFF/GTF) file they use. From their results, this reviewer believes that they have used StringeTie, or similar software. The authors need to detail precisely the pipeline of RNA-Seq analyses performed.
Heatmaps should be done in a colour that is distinguishable by colour blind people, preferably red and blue, or one of the following (https://github.com/wistia/heatmap-palette)
From the point of view of this reviewer, the presented MS needs to correct and clarify all the indicated methodological points, as well as the correct reannotation of genes.
Author Response
Please see the attachment.
Response to Reviewer 1 Comments
|
||
1. Summary |
|
|
We sincerely thank you for taking your valuable time and energy to review this manuscript. According to your professional and constructive comments, we have revised the manuscript point-to-point in accordance with your valuable suggestions, and the corresponding corrections were marked as red in the re-submitted files. The revised information and responses are as follows. |
||
2. Questions for General Evaluation |
Reviewer’s Evaluation |
Response and Revisions |
Does the introduction provide sufficient background and include all relevant references? |
Can be improved |
|
Are all the cited references relevant to the research? |
Yes |
|
Is the research design appropriate? |
Must be improved |
|
Are the methods adequately described? |
Must be improved |
|
Are the results clearly presented? |
Must be improved |
|
Are the conclusions supported by the results? |
Must be improved |
|
3. Point-by-point response to Comments and Suggestions for Authors |
||
Comments 1: The manuscript entitled “Transcriptome Sequencing Reveals the Mechanism of Auxin Regulation of Root Expansion in Carrot” by Li and collaborators presents present a transcriptomic and metabolomic analysis of the agricultural species Daucus carota. The approach and the analyses present in the MS are of interest, as well as the intention to describe the role of a very relevant hormone in the root such as auxins. However, it is necessary that the authors detail more precisely the genetic annotation, since carrot is not a model species, and it is also necessary that their experimental system really reflects the auxin dynamics in an organism as complex as the carrot root. The authors should take the following major points into account On the experimental system. The authors only analyse the gene expression and meatabolomics of one region of the root, line 338 "middle part of carrot roots with similar growth". It is necessary for the authors to demonstrate that this region is indeed representative of the whole root. In their results and conclusions, they assume that what happens in this region can be extrapolated to the rest of the whole organ. In fact, auxins are a hormone that undergoes extensive movement throughout the plant, and in the root it has a polar transport. For this reason, it is necessary to show that this region is representative of the whole root. |
||
Response 1: Thank you for pointing this out. Because we mainly study root enlargement, we think the middle part is more representative. |
||
Comments 2: As for gene annotation, when working with a non-model organism, such as Arabidopsis thaliana, it is necessary to provide a lot of evidence for gene annotation. The authors do not indicate how they have identified the genes, how they have annotated them and how they have identified the different genes associated with auxin biosynthesis, metabolism and signal transduction. This point should be well explained in the materials and methods section. In fact, there are incorrect annotations of the genes, as an example; DCAR_013008 annotated as TAA1 by the authors, appears as orthologue of the FLOE1 gene (AT4G28300) in Esemble database (https://plants.ensembl.org/Daucus_carota/Gene/Compara_Ortholog?db=core;g=DCAR_013008;r=4:33572028-33577084;t=KZM99630). The same result is obtained with another gene also identified as TAA1, DCAR_021243, in this case through a phylogeny also from Emsembl database (https://plants.ensembl.org/Daucus_carota/Gene/Compara_Tree?anc=4697284;db=core;g=DCAR_021243;g1=AT4G28300;r=6:24748057-24751437;t=KZM91392). These are just two examples where the annotations or associations made by the authors do not match those present in the databases, and the same is true when checked in the Plaza Dicots database (https://bioinformatics.psb.ugent.be/plaza/versions/plaza_v5_dicots/). Authors should indicate and correct the annotations they indicate. In the case of complex families such as ARF and YUCCA, the construction of phylogenetic trees is also necessary. |
||
Response 2: Agree. We have added the details in 4.3.2 and 4.4 of Materials and Methods in lines 397 - 399 of page 17 and lines 414 - 416 of page 18, respectively, and marked them as red. We have checked all the genes, deleted some genes that did not match the Arabidopsis homologous gene annotations, and marked them as red in Table 1 on pages 6 and 7, Table 2 on page 9, Table 3 on page 11, and Table 4 on page 13. The phylogenetic tree of YUCCA and ARF have been constructed in the published papers, respectively, and we cited them in line 169 of page 5 and line 239 of page 10 in this paper ([17] Yan et al., 2021; [21] Pei et al., 2021). |
||
Comments 3: Finally, regarding the RNA-Seq analysis, the authors only indicate which software they use for mapping, in this case HISAT2 (Line 365), but they do not detail how they do the gene counting, and they do not indicate which gene structural prediction (GFF/GTF) file they use. From their results, this reviewer believes that they have used StringeTie, or similar software. The authors need to detail precisely the pipeline of RNA-Seq analyses performed. |
||
Response 3: Agree. We have added the details in 4.3.2 of Materials and Methods in lines 396 - 397 of page 17 and marked them as red. |
||
Comments 4: Heatmaps should be done in a colour that is distinguishable by colour blind people, preferably red and blue, or one of the following (https://github.com/wistia/heatmap-palette). |
||
Response 4: Agree. We have modified the heatmap to red and blue (Figure 7 on page 14). |
||
Comments 5: From the point of view of this reviewer, the presented MS needs to correct and clarify all the indicated methodological points, as well as the correct reannotation of genes. |
||
Response 5: We have added the methods in detail in the Materials and Methods part and re-annotated some genes. |
||
4. Response to Comments on the Quality of English Language |
||
Point 1: I am not qualified to assess the quality of English in this paper. |
||
Response 1: We have entrusted the MDPI language editing service to polish the manuscript and marked them as red in the manuscript. |
||
5. Additional clarifications |
||
We have nothing else to elaborate on and thank you again for your valuable comments. |

Reviewer 2 Report
Comments and Suggestions for Authors
Review ijms 2792152
The manuscript by Li and colleagues entitled “Transcriptome Sequencing Reveals the Mechanism of Auxin Regulation of Root Expansion in Carrot” contains data on the expression of IAA synthesis, degradation and signalling genes, as well as IAA content at different growth stages of carrot root. The authors revealed 12 IAA synthesis, 8 IAA degradation and 13 IAA signalling genes which were differentially expressed during root expansion. Expression values of 11 genes significantly correlated with the changes in IAA content suggesting that they play a crucial role in IAA regulation. Unlike Cruciferae, the IAOx pathway was not involved in IAA synthesis in carrot in which the tryptamine, IPyA, and IAM pathways were dominant. The information obtained on the content of IAA and the regulation of the expression of genes encoding it during carrot ontogenesis is of certain interest.
However, despite the authors' claims that the changes of IAA content are consistent with the mechanisms of regulation of root expansion in carrot, the manuscript does not contain transcriptomic data on the expression on the genes directly related to cell proliferation and expansion, or even data on the correlation of the given growth indicators with the IAA content.
I would recommend revising the concept of the article, focusing on changes in the content of auxins during the growth and development of carrot roots. In this case it would be desirable to supplement the work with data on the detailed content of auxins and their precursors and metabolites, linking these data with the expression of auxin encoding genes.
If the authors claim to reveal the mechanisms of regulation of root growth by IAA, they should include an analysis of the relationship between hormonal data and morphological indices as well as the expression of genes directly guiding root growth. The manuscript must contain chapter “Screening of Expansion-related Genes Regulated by IAA” (Line 381).
Minor concerns:
Lines 112-139. The description of IAA metabolism process would be more appropriate in the Introduction, especially since the necessary information is duplicated in the description of the results and even in the Materials and Methods (lines 372-380)
The manuscript needs careful editing by a native speaker, which will contribute to its better perception by readers
Comments on the Quality of English Language
Extensive editing of English language required
Author Response
Please see the attachment.
Response to Reviewer 2 Comments
|
||
1. Summary |
|
|
We sincerely thank you for taking your valuable time and energy to review this manuscript. According to your professional and constructive comments, we have revised the manuscript point-to-point in accordance with your valuable suggestions, and the corresponding corrections were marked as red in the re-submitted files. The revised information and responses are as follows. |
||
2. Questions for General Evaluation |
Reviewer’s Evaluation |
Response and Revisions |
Does the introduction provide sufficient background and include all relevant references? |
Can be improved |
|
Are all the cited references relevant to the research? |
Yes |
|
Is the research design appropriate? |
Must be improved |
|
Are the methods adequately described? |
Can be improved |
|
Are the results clearly presented? |
Can be improved |
|
Are the conclusions supported by the results? |
Not applicable |
We have supplemented the correlation analysis between morphological indicators and hormone content in the manuscript. It will help to support the conclusion. |
3. Point-by-point response to Comments and Suggestions for Authors |
||
Comments 1: The manuscript by Li and colleagues entitled “Transcriptome Sequencing Reveals the Mechanism of Auxin Regulation of Root Expansion in Carrot” contains data on the expression of IAA synthesis, degradation and signalling genes, as well as IAA content at different growth stages of carrot root. The authors revealed 12 IAA synthesis, 8 IAA degradation and 13 IAA signalling genes which were differentially expressed during root expansion. Expression values of 11 genes significantly correlated with the changes in IAA content suggesting that they play a crucial role in IAA regulation. Unlike Cruciferae, the IAOx pathway was not involved in IAA synthesis in carrot in which the tryptamine, IPyA, and IAM pathways were dominant. The information obtained on the content of IAA and the regulation of the expression of genes encoding it during carrot ontogenesis is of certain interest. However, despite the authors' claims that the changes of IAA content are consistent with the mechanisms of regulation of root expansion in carrot, the manuscript does not contain transcriptomic data on the expression on the genes directly related to cell proliferation and expansion, or even data on the correlation of the given growth indicators with the IAA content. I would recommend revising the concept of the article, focusing on changes in the content of auxins during the growth and development of carrot roots. In this case it would be desirable to supplement the work with data on the detailed content of auxins and their precursors and metabolites, linking these data with the expression of auxin encoding genes. If the authors claim to reveal the mechanisms of regulation of root growth by IAA, they should include an analysis of the relationship between hormonal data and morphological indices as well as the expression of genes directly guiding root growth. The manuscript must contain chapter “Screening of Expansion-related Genes Regulated by IAA” (Line 381). |
||
Response 1: In this study, we did not measure the content of auxin precursors and metabolites and therefore cannot supplement this data. We have supplemented the correlation analysis between morphological indicators and hormone content in lines 83 - 92 of the results section. In addition, we added the correlation analysis figure as Figure 2B on page 3. However, we wonder if it's appropriate. We cannot find genes related to root enlargement in our differentially expressed gene annotation files, therefore we cannot add the chapter “Screening of Expansion-related Genes Regulated by IAA”. |
||
Comments 2: Lines 112-139. The description of IAA metabolism process would be more appropriate in the Introduction, especially since the necessary information is duplicated in the description of the results and even in the Materials and Methods (lines 372-380). |
||
Response 2: We think the content of the description of auxin metabolism is not necessary in the introduction, however, if not stated in the results and analysis, it would affect the reader's understanding of the content, so we hope to retain it there. We have deleted the duplicated content in the Materials and Methods in lines 406 - 409 of page 18 and marked them as red. |
||
Comments 3: The manuscript needs careful editing by a native speaker, which will contribute to its better perception by readers. |
||
Response 3: We will entrust the MDPI language editing service to polish the manuscript. |
||
4. Response to Comments on the Quality of English Language |
||
Point 1: Extensive editing of English language required |
||
Response 1: We have entrusted the MDPI language editing service to polish the manuscript and marked them as red in the manuscript. |
||
5. Additional clarifications |
||
We have nothing else to elaborate on and thank you again for your valuable comments. |

Round 2
Reviewer 1 Report
Comments and Suggestions for Authors
The manuscript entitled “Transcriptome Sequencing Reveals the Mechanism of Auxin Regulation of Root Expansion in Carrot” by Li and collaborators present a new version of their manuscript, which presents improvements in accordance with the reviewers' comments. However, this new MS still presents different points that have not been clarified or improved by the authors. In this sense, this reviewer thinks that the authors should take these major points into account:
1.- The authors have not clarified or demonstrated that the carrot root fragment they use for their analysis is representative of the whole organ. They should provide experimental evidence that the fragments used represent the whole organelle and are representative of the auxin dynamics they describe.
2.- The annotation of the genes has not been corrected, they have only been limited to the genes that this reviewer has given as an example. The annotation of the tables has not been revised in depth, in addition, there are references that the authors cite that are impossible to locate. It is not possible to generate conclusions from genes that are not correctly annotated. As an example, the gene DCAR_004140, in Table 3, annotated as TIR1, which is not orthologous with A. thaliana in the Emsembl database. An example of a citation that cannot be located is [17] Yan et al., 2021.According to this reviewer, the authors should check all the genes in the tables they present in the Emsembl database, not by blast which only identifies homologues not orthologues, in the orthologues section of this database and confirm them in other databases such as PantherDB (https://www.pantherdb.org/) or PlazaDicoits. Also, provide phylogenetic trees, for complex families, such as those of AAIs. If you cite previous studies, which are not available to the whole scientific community, you should include the phylogenetic trees as supplementary and indicate the study they come from.
As in my previous review, the gene annotation is not correct, as this reviewer has found discrepancies between those presented by the authors and different curated databases. This is a point that needs to be clarified, as most of the study is based on genomic annotation.
3.- Regarding the RNA-Seq analysis, it is true that the authors have improved the description of the analysis, but they only indicate the databases used, not the tools they have used to rasterise these databases. It remains unclear how they have done the analysis. For example, StringeTie generates beads and the authors use FPKM as a measure of gene expression, how do they do this normalisation? The RNA-Seq analysis is still unclear.
Figure 6 is not a correlation, it is simply the heat maps, the authors should make a correct correlation, such as Pearson or similar, as well as give a statistic of this analysis such as p-value or similar.
Minor changes
Figure 1. Images should be removed from the rulers, for a bar representing the size. The image needs to be reworked, it is too raw.
Figure 2 and 3 can be merged into a single panel.
Overall, this reviewer thinks the study is interesting, but the authors should contribute and improve the genomic annotation. The authors should also provide supplementary tables with the information collected from the annotation in the different databases suggested by this reviewer. When studying a non-model organism, it is necessary to correctly annotate the genes using different databases, in order to have a strong support on the results obtained.
Reviewer 2 Report
Comments and Suggestions for Authors
Review 2
The authors significantly improved the quality of the English language of the manuscript. They also excluded repetitions from the text and introduced correlation analysis data between morphological indicators and hormone content. However, the results of “Screening of Expansion-rRelated Genes Regulated by IAA”, stated in the chapter “Materials and methods“ (Lines 406-409) are still missing. Moreover, the authors acknowledge that they “cannot find genes related to root enlargement in our differentially expressed gene annotation files, therefore we cannot add the chapter “Screening of Expansion-related Genes Regulated by IAA”. Without these data, the authors' claims to identify the mechanisms of “Auxin Regulation of Root Expansion in Carrot” remain unfounded. Analysis of correlations between IAA-related DEGs and IAA levels shows that these genes are only closely associated with IAA levels at certain stages of carrot root development.
The manuscript still needs significant revision.